# LAMM: Language-Assisted Multi-Modal Instruction-Tuning Dataset, Framework, and Benchmark

**Zhenfei Yin**[1,3]*, **Jiong Wang**[1,4]*, **Jianjian Cao**[1,4]*, **Zhelun Shi**[1,2]*, **Dingning Liu**[1,5], **Mukai Li**[1]
**Xiaoshui Huang**[1], **Zhiyong Wang**[3], **Lu Sheng**[2], **Lei Bai**[1]†, **Jing Shao**[1]†, **Wanli Ouyang**[1]
[1]Shanghai Artificial Intelligence Laboratory
[2]Beihang University [3]The University of Sydney [4]Fudan University [5]Dalian University of Technology
`{yinzhenfei,bailei,shaojing}@pjlab.org.cn`

## Abstract

Large language models have emerged as a promising approach towards achieving general-purpose AI agents. The thriving open-source LLM community has greatly accelerated the development of agents that support human-machine dialogue interaction through natural language processing. However, human interaction with the world extends beyond only text as a modality, and other modalities such as vision are also crucial. Recent works on multi-modal large language models, such as GPT-4V and Bard, have demonstrated their effectiveness in handling visual modalities. However, the transparency of these works is limited and insufficient to support academic research. To the best of our knowledge, we present one of the very first open-source endeavors in the field, LAMM, encompassing a Language-Assisted Multi-Modal instruction tuning dataset, framework, and benchmark. Our aim is to establish LAMM as a growing ecosystem for training and evaluating MLLMs, with a specific focus on facilitating AI agents capable of bridging the gap between ideas and execution, thereby enabling seamless human-AI interaction. Our main contribution is three-fold: 1) We present a comprehensive dataset and benchmark, which cover a wide range of vision tasks for 2D and 3D vision. Extensive experiments validate the effectiveness of our dataset and benchmark. 2) We outline the detailed methodology of constructing multi-modal instruction tuning datasets and benchmarks for MLLMs, enabling rapid scaling and extension of MLLM research to diverse domains, tasks, and modalities. 3) We provide a primary but potential MLLM training framework optimized for modality extension. We also provide baseline models, comprehensive experimental observations, and analysis to accelerate future research. Our baseline model is trained within 24 A100 GPU hours, framework supports training with V100 and RTX3090 is available thanks to the open-source society. Codes and data are now available at https://openlamm.github.io/.

## 1 Introduction

Humans interact with the real world through multi-modal information, such as vision and language, since each modality possesses unique capabilities to describe the world, thereby providing us with richer information to construct our world model. Developing AI agents capable of processing such multi-modal information, learning and memorizing world knowledge from it, and comprehending

---

*Equal Contribution
†Corresponding Authors: Jing Shao (shaojing@pjlab.org.cn) and Lei Bai (bailei@pjlab.org.cn)

37th Conference on Neural Information Processing Systems (NeurIPS 2023) Track on Datasets and Benchmarks.

open-world instructions from humans to take actions and complete complex tasks has long been a core aspiration in artificial intelligence.

Large Language Models (LLM) have made remarkable progress toward achieving that aspiration. ChatGPT and GPT-4 [1] model can directly comprehend user intents and generalize to unknown real-world tasks [2]. LLM has become a universal task interface for general purposes. Almost all natural language understanding and generation tasks can be transformed into instruction inputs, enabling a single LLM to perform zero-shot generalization on various downstream applications [3, 4, 5]. Within the realm of open-source models, the LLaMA series [6, 7] stands out for its performance and transparency. Building upon the LLaMA ecosystem, models like Alpaca [8] and Vicuna [9] employ different strategies, such as utilizing various machine-generated high-quality instruction-following samples, to enhance the performance of LLMs, showcasing impressive results. Notably, these efforts are all text-only. While Multi-model Large Language Models (MLLM) like GPT-4V [10] and Bard [10] demonstrate remarkable capabilities in processing visual inputs, unfortunately, they are not currently available for use within the open-source academic community.

Hence, we present LAMM, encompassing the Language-Assisted Multi-Modal instruction tuning dataset, framework, and benchmark. As one of the very first open-source endeavors in MLLMs, our aim is to establish LAMM as a thriving ecosystem for training and evaluating MLLMs, and further empower us to cultivate multi-modal AI agents capable of bridging the gap between ideas and execution, facilitating seamless interaction between humans and AI machines. In this work, LLMs serve as the universal task interface, with inputs from vision tokens provided by pre-trained multi-modal encoders and language instructions. The powerful modeling capability of LLMs, combined with a unified optimization objective, can help align the model to various modalities. This design sets LAMM apart from visual foundation models [11, 12], where each model is finely tuned for a specific task, or from multi-modal visual language foundation models that can only be used as pre-trained models for visual tasks or possess limited zero-shot capabilities [13], or from multi-task foundation models struggle in tag-of-war problems [14].

Thoroughly, we present a novel instruction tuning dataset, which extends the research of MLLMs to both image and point cloud. Our dataset emphasizes fine-grained information and factual knowledge. Additionally, we introduce the very first attempt of a benchmark for MLLMs that offers a comprehensive evaluation of existing open-source models on various computer vision tasks, with two new evaluation strategies designed explicitly for multi-modal language models. We conduct over 200 experiments to provide extensive results and valuable observations on the capabilities and limitations of MLLMs. Also, we establish an extensible framework to facilitate the extension of multi-modal language models to additional modalities. Our baseline model surpasses existing multi-modal language models in downstream tasks related to images, demonstrating the effectiveness of our framework and dataset. Above all, we have open-sourced our complete codebase for training and evaluating MLLMs, instruction tuning dataset covering both image and point cloud. various baseline models trained with our dataset and framework utilizing different settings to promote the development of an open research community for MLLMs.

**Dataset** We include an image instruction tuning dataset containing 186,098 image-language instruction-response pairs and a point cloud instruction tuning dataset with 10,262 point cloud-language instruction-response pairs. Motivated by LLaVA [15] and GPT-4V [10], we collect images and point clouds from publicly available datasets and use the GPT-API through self-instruction [16] methods to generate instructions and responses based on the original labels from these datasets. The resulting dataset has three appealing properties: 1) To emphasize fine-grained and dense information, we add more visual information, such as visual relationships and fine-grained categories as input for the GPT-API. 2) We observe on our benchmark that existing MLLMs may struggle to understand vision task instructions. To address this, we designed a method to convert vision task annotations into instruction-response pairs, which enhances MLLMs' understanding and generalization of vision task instructions. 3) Considering the vulnerability of LLMs to the hallucination on factual knowledge, our dataset also includes data pairs for commonsense knowledge question answering by incorporating a hierarchical knowledge graph label system from the Bamboo [17] dataset and the corresponding Wikipedia description.

**Benchmark** We evaluate 9 common image tasks, using a total of 11 datasets with over 62,439 samples, and 3 common point cloud tasks, by utilizing 3 datasets with over 12,788 data samples, while existing works only provide quantitative results on fine-tuning and evaluating specific datasets such

as ScienceQA, and most works only conduct demonstration or user studies. 1) We are the very first attempt to establish a benchmark for MLLMs. We conducted a comprehensive benchmark to quantify the zero-shot and fine-tuning performance of existing multi-modal language models on various computer vision tasks and compare them against state-of-the-art methods of these tasks, including classification, object detection, pose estimation, visual question answering, facial classification, optical character recognition, object counting. 2) We also attempted two novel evaluation strategies designed explicitly for MLLMs. Specifically, as for language performance on text generation, we established a scoring logic based on the GPT-API. And for tasks involving interactions between localization points and query images, such as object detection and pose estimation, we proposed an object-locating evaluation method.

**Framework** To validate the effectiveness of our dataset, we propose a primary but potential MLLM training framework. To avoid modality conflicts caused by introducing multiple modalities, we differentiate the encoder, projector, and LLM finetuning blocks for different modalities in the framework design. Meanwhile, by adding encoders and decoders for other modalities, our framework can flexibly extend to cover more modalities and tasks, such as video understanding, image synthesis, and so on. We provide the results of our baseline models trained using this framework on our benchmark and present various observations to accelerate future research.

## 2 Related Work

**Multimodal Large Language Model.** With the rapid development of Large Language Models (LLM) such as ChatGPT, GPT-4 [1], many studies manage to explore incorporating other modalities based on LLM and they can be categorized into two perspectives. **1) System Design Perspective:** Visual ChatGPT [18] and MMREACT [19] invoke various vision foundation models by processing user query to investigate the visual roles of ChatGPT with the help of Visual Foundation Models. ViperGPT [20] instructs LLM to parse visual queries into interpretable steps expressed by Python code. HuggingGPT [21] extends its framework to more modalities by integrating more expert models on Huggingface. **2) End-to-End Trainable Model Perspective:** The other methodology is to connect models for different modalities into an end-to-end trainable model, also known as multimodal large language model. Flamingo [22] proposes a unified architecture for language and vision modeling, while BLIP-2 [23] introduces a Querying Transformer to connect information from image to text modality. Kosmos [4] and PaLM-E [24] build an end-to-end trainable framework on web-scale multi-modal corpora. With the open-sourced LLaMA [6], Mini-GPT4 [25] optimizes a trainable projection matrix only, which connects pre-trained BLIP-2 style vision encoder and large language model, while LLaVA [15] and mPLUG-OwL [26] also finetune LLM. Besides feeding visual info to LLM as input only, LLaMA-Adapter [27], Multi-modal GPT [28] and Otter [29] also integrate multi modal information with intermediate features in LLM.

**Instruction Tuning.** Instruction tuning [30] is a method proposed to improve the ability of large language models to follow instructions and enhance downstream task performance. Instruction-tuned models like InstructGPT [31], OPT-IML [32], Alpaca [8], have shown promising improvement compared to their based model. The existing instruction tuning datasets are primarily derived from collections of academic datasets like FLAN [30], chatbot data collected from ChatGPT usage such as ShareGPT, or constructed using self-instruction [16] methods like Alpaca. Apart from pure text instruction tuning datasets, Multi-Instruct [33] covers 47 multi-modal tasks. Mini-GPT4 [25] constructs instruction following dataset by composing image-text datasets and handwritten instruction templates. Moreover, LLaVA [15] feeds captions and bounding boxes as the context of COCO images to GPT-4 and therefore get 150K instruction data. Otter [29] builds such instruction tuning datasets from multi-modal MMC4 dataset [34] and incorporates in-contextual examples into instruction tuning by grouping similar instructions together.

## 3 Dataset

We introduce a comprehensive multi-modal instruction tuning dataset, which involves images and point clouds from publicly available datasets for diverse vision tasks, as well as high-quality instructions and responses based on the GPT-API and self-instruction methods [16]. To be specific, our dataset contains 186K language-image instruction-response pairs, and 10K lanuage-3D instruction-response pairs. Figure 1 provides an overview of its construction process. We provide detailed information on how to construct the multi-modal instruction tuning dataset to guide the academic community, facilitating the replication and further development of our work. We showcase additional demonstrations of sample data and provide a complete prompting method in the Appendix.

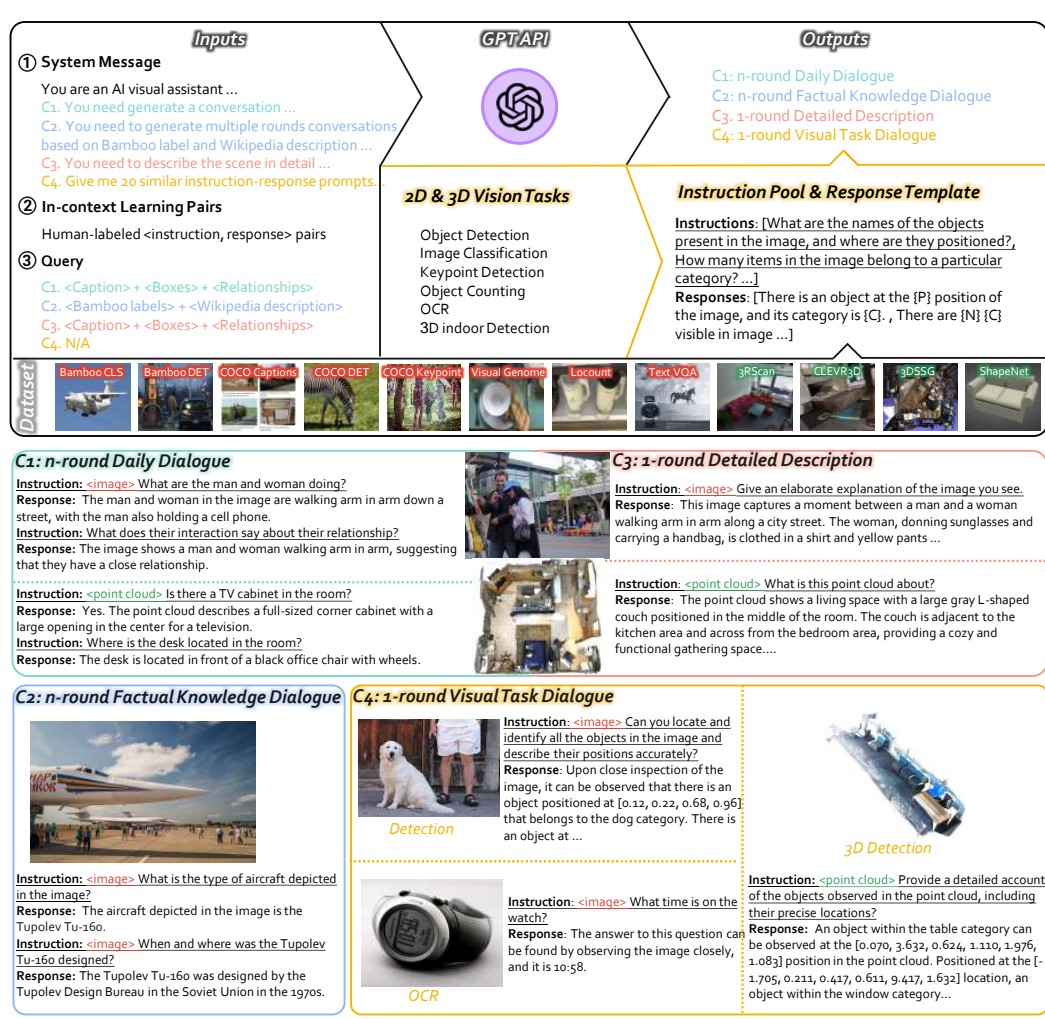

Figure 1: Overview of our dataset, demonstrating the process of constructing our Instruction Tuning dataset using the GPT-API. By designing different system messages, in-context learning pairs, and queries, we have created the dataset that covers almost all high-level vision tasks for both 2D and 3D vision. The dataset includes four distinct groups: n-round Daily Dialogue, n-round Factual Knowledge Dialogue, 1-round Detailed Description, and 1-round Visual Dialogue. It is worth noting that for the introduction of vision tasks, we only used the GPT-API to generate instruction-response templates and did not directly generate dialogue data. Finally, some examples of the dataset are presented below, including 2D and 3D scenes and their corresponding instruction-response pairs.

We design four kinds of multi-modal instruction-response pairs: 1) *C1: n-round daily dialogue* focuses on multi-modal daily conversations. 2) *C2: n-round factual knowledge dialogue* aims at dialogues requiring factual knowledge reasoning. 3) *C3: 1-round detailed description* aims to elaborate images and 3D scenes in texts. 4) *C4: 1-round visual task dialogue* transfers vision tasks into instruction-response pairs, aiming at enhancing generalization ability towards visual tasks.

We include diverse 2D and 3D vision tasks into the dataset, such as captioning, scene graph recognition and VQA that are directly compatible with natural languages, as well as classification, detection, counting and OCR that output labels, bounding boxes, digits and a list of words instead. Note that the point-cloud instruction tuning dataset does not include data in the *C2: n-round factual knowledge dialogue* category. This is due to the current lack of publicly available 3D datasets with a well-defined labeling system containing factual knowledge. In our dataset, the instruction-response pairs are gathered from 8 image datasets and 4 point cloud datasets, which are referred in Figure 1.

The first three types of instruction-response pairs are generated by inputting several special designed prompts to the GPT-API, namely *system messages*, *in-context learning pairs* and *queries*: (1) *System messages* are to inform the GPT-API about the task definitions and requirements. (2) Several *in-*

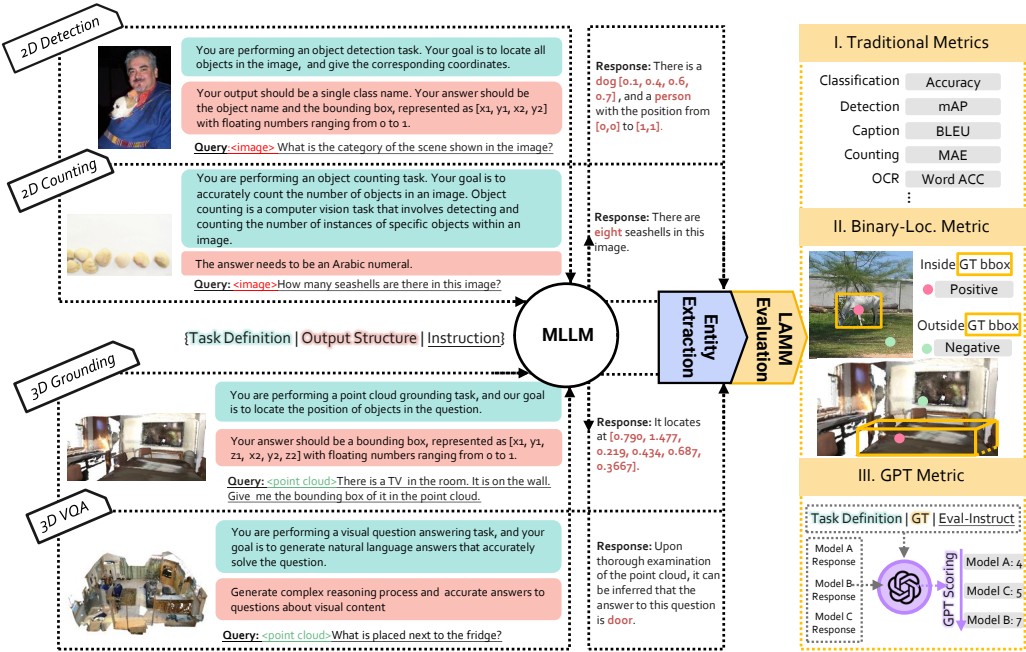

Figure 2: An overview of our Benchmark. It includes both 2D and 3D pipelines, covering multiple computer vision tasks. For each task, we provide the task definition, output structure, and a set of questions as instructions to the MLLM model. Then the entity extraction is applied on the output to extract the key answer. The LAMM Evaluation is used to evaluate the model's performance, which includes traditional metrics, binary-location metric and the GPT Metric.

*context learning pairs* are manually annotated to ensure that the rest instruction-response pairs can be generated by a similar fashion. (3) *Queries* include comprehensive annotations of captions, bounding boxes of objects, relations between objects, factual knowledges from the Bamboo's label system and their Wikipedia descriptions.

The last type of instruction-response pairs also apply the system messages and in-context learning pairs, but use GPT-API to generate a pool of templates of instruction-response pairs instead. In this way, ground-truth annotations of many vision tasks, such as object/keypoint detection, OCR, counting and *etc*., can be inserted into these templates, and thus are easier to be converted into reliable language responses, rather than aforementioned query-based conversion.

## 4 Benchmark

Different from LLaVA [15], MiniGPT4 [25] and mPLUG-owl [26] that only provide demos and user studies to qualitatively evaluate the performances of their MLLMs, we propose the first benchmark of MLLMs, which instead evaluates the quantitative performance of MLLMs on various 2D and 3D vision tasks. It includes an inference pipeline and a set of evaluation metrics. To be specific, the benchmark on 2D vision tasks evaluates 9 common image tasks, using a total of 11 datasets with over 62,439 samples. The benchmark on 3D vision tasks evaluates 3 common point cloud tasks, by utilizing 3 datasets with over 12,788 data samples.

**Inference Pipeline.** It ensures that the MLLMs can produce reasonable responses that can be fairly evaluated, which includes the way of processing input instructions and the extracting output entities. We construct the *Inference Instruction* to help the model better understand the task it is performing and the output structure that is required, aim to improve the stability and reliability of the benchmarking process. Inference Instruction includes Task Definition, Output Structure and the usually employed Query Questions, as shown in Figure 2. Inspired by chain-of-thought prompting methods [35], we also prompt the MLLM to perform complex reasoning followed by the final answer, so as to obtain a more reliable answer. Then, we employ the Natural Language Toolkit (NLTK) and regular expression matching to extract entities from the output text. These entities act as the results.

**Evaluation Metrics.** The set of evaluation metrics includes Traditional Metrics, Binary Locating Metric, and GPT Metric. The Traditional Metrics are task-specific metrics from the listed 2D and 3D

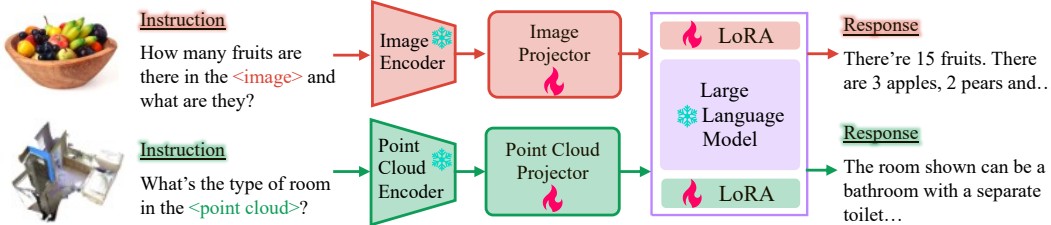

Figure 3: Framework of multi-modality language model. Each modality is encoded by corresponding pre-trained encoder and decoded by LLM. LLM is shared among modalities and trainable projection layers and LoRA parameters are modality-specific.

vision tasks, which are the most rigorous to evaluate how MLLMs handle vision tasks. In the Binary Locating Metric, the model needs to output an approximated location of a recognized object through the instruction "output the position of the object", whose result is considered true if it is within the object's groundtruth bounding box. It is a straightforward metric to compare the localization ability of an MLLM model. To evaluate the understanding and question-answering abilitis of MLLM models, we utilize the GPT metric to evaluate the answers' relevance and accuracy to the groundtruth. To be specific, we prompt GPT to assign scores to the outputs generated by each model through the instruction described in Figure 2. The scoring criteria were based on accuracy, relevance, fluency, logical coherence, and information richness.

**Evaluation Settings.** All 2D and 3D vision tasks can be evaluated in a zero-shot manner, where the testing data have no intersection with MLLM's training data. Moreover, we also evaluate the finetuning ability of MLLMs on the test dataset about several mainstream tasks, such as detection, classification and VQA in 2D tasks, as well as detection, grounding and VQA in 3D tasks.

## 5 Experiments and Results

### 5.1 Framework

The overall framework of our baseline MLLM is depicted in Figure 3. Each modality, image or point cloud, is processed by corresponding encoder, whose features are then projected to the same feature space as the text embeddings by a trainable projection layer. Instructions are directly tokenized by SentencePiece tokenizer [36], then the vision and text tokens are concatenated to feed into the LLM model. To finetune LLM efficiently, we add LoRA [37] parameters to all projection layers in the self-attention layers. LoRA parameters for different vision modalities are not shared. Multi-modal tokens are decoded by a shared LLM model and the corresponding LoRA parameters. As shown in Figure 3, only feature projectors and LoRA parameters are optimized during training. We use Vicuna-13B [9], as our LLM. Rank of LoRA modules are set to 32. We train all parameters including projection layers and LoRA modules in a one-stage end-to-end fashion with 4 A100 GPUs.

Input images are resized to be 224×224 and split into 256 patches. We use CLIP [38] pre-trained ViT-L/14 and use image patch features output from transformer layers as image representations. We follow the design of FrozenCLIP [39] to encode point clouds, in which point cloud is tokenized to be 256 tokens by PointNet++ [40] and further encoded by CLIP pretrained ViT-L/14.

### 5.2 Results on Traditional Metrics

**Zero-shot Setting on 2D Vision Tasks.** Table 1 shows the results of MLLM on 2D vision tasks by the Traditional Metrics. All the MLLM models were tested in a zero-shot setting. Although MLLM models demonstrated certain abilities of recognizing open-vocabulary classes, understanding images, and answering questions, they performed poorly on tasks involving object localization, including object detection, counting and keypoints detection. *Localization-aware Tasks:* In detection tasks, our baseline model demonstrated stronger localization ability, but there is still a significant gap between the predicted and the ground-truth bounding boxes, indicating MLLMs' weakness to output certain digits representing points and reasoning spatial information. In counting tasks, the MLLM models showed a significant gap between the predicted and ground truth number of objects. MiniGPT4 failed in this task as it is unable to provide a specific number for most of the

Table 1: Comparison of Multi-modal Large Language Models on 2D vision tasks.

| Task | Dataset | Metric | LLaVA[15] | MiniGPT4[25] | mPLUG-owl[26] | LAMM |
|---|---|---|---|---|---|---|
| Classification | CIFAR10 [41] | Acc ↑ | **60.83** | 46.22 | 42.5 | 37.9 |
| Detection | VOC2012 [42] | mAP ↑ | 1.42 | 0.92 | 0.158 | **7.20** |
| VQA | SQAimage [43] | Acc ↑ | 40.5 | 43.43 | 36.39 | **49.88** |
| | AI2D [44] | | 18.13 | Failed | 19.31 | **20.92** |
| Image Caption | flickr30k [45] | BLEU4 ↑ | **6.65** | 5.1 | 2.74 | 2.56 |
| F-g classification | UCMerced [46] | Acc ↑ | **47** | 33.6 | 32.5 | 18.23 |
| Counting | FSC147 [47] | MAE ↓ | 56.2 | Failed | 60.67 | **46.88** |
| OCR | SVT [48] | Word Acc ↑ | **37.78** | 16.97 | 30.39 | 29.14 |
| Facial Classification | CelebA(Smile) [49] | Acc ↑ | Failed | **66.36** | Failed | 57.50 |
| | CelebA(Hair) [49] | | **46.42** | 43.47 | 40.93 | 56.96 |
| Keypoints Detection | LSP [50] | PCK ↑ | Failed | Failed | Failed | Failed |

Table 2: Results of our baseline model on selected 2D vision tasks. Both zero-shot test result and finetuned results reported. Metrics for classification and VQA is **accuracy**, and that for object detection is **mAP@0.5**.

| Task | Dataset | LAMM (Zero-Shot) | LAMM (Finetune) |
|---|---|---|---|
| Classification | CIFAR10 [41] | 37.9 | 91.2 |
| Object Detection | VOC2012 [42] | 7.20 | 13.48 |
| VQA | SQAimage [43] | 49.88 | 74.27 |

data. As for the keypoints detection task, we asked the MLLM models to predict the position of each human keypoint in turn. However, all the predicted positions were not in an acceptable range. The MLLMs show a significant gap in this task, indicating that they have difficulty in accurately predicting the locations of the keypoints. *VQA Tasks:* Our baseline model demonstrated certain advantages in image understanding and multiple-choice question answering compared to other models. Note that the LLaVA model we compared to was evaluated in the zero-shot setting. Additionally, we removed the random choice process from the LLaVA evaluation to obtain a more straightforward evaluation. *Captioning Tasks:* All MLLM models performed poorly on image captioning. We argue that BLEU4 is not an appropriate metric since longer captions may lead to lower scores, and MLLMs tend to output detailed description. *Classification Tasks:* In fine-grained classification tasks and face classification tasks, all MLLMs performed poorly. Specifically, on the CelebA (Smile) dataset, the LLaVA model outputs "yes" to all the queries, while the mPLUG model randomly gives predictions. However, regarding the CelebA (Hair) dataset, the MLLMs can recognize hair color since the ability to infer visual knowledge for color recognition is relatively straightforward. These results suggest that the MLLM models may have difficulty in tasks that require fine-grained distinctions. *OCR Tasks:* As for OCR tasks, LLaVA can recognize and extract text from images. However, our baseline model performed poorly on this task. We provide more analysis of the results and identify several potential reasons for the performance gap in the Appendix.

**Fine-tuning Setting on Image Tasks.** We also fine-tuned our baseline model on several vision datasets, including CIFAR10, VOC2012, and SQAimage. The results are shown in Table 2. The fine-tuned baseline achieved an accuracy of 91% on CIFAR10. It also achieved an mAP of 13% on VOC2012, in comparison with 4.8% in the zero-shot setting. These results indicate that our baseline models can receive the ability of localizing objects after being fine-tuned on detection data.

**Zero-shot Setting on Point Cloud Tasks.** Table 3 shows the result of our baseline model on 3D scene understanding tasks, under the zero-shot and fine-tuning settings, respectively. The results after finetuning are significantly better than the zero-shot setting, in all test tasks. Our baseline model finetuned on ScanQA multiple choice data almost achieves 100% accuracy, which may have an overfitting issue due to the narrow training/test gap and small scale of 3D dataset.

## 5.3 Results of Binary Locating Metric and GPT Metric

**Binary Locating Metric.** Table 4 shows the zero-shot results of the MLLMs on the proposed Binary Locating Metric and GPT Metric. The Binary Locating Metric covers the data from VOC2012,

Table 3: Results of 3D tasks. Metrics for 3D object detection and visual grounding is **mAP@0.5**, and that for 3D VQA is **accuracy** of multiple choice problem.

| Task | Dataset | LAMM (Zero-Shot) | LAMM (Finetune) |
|---|---|---|---|
| 3D Object Detection | ScanNet[51] | 9.3 | 11.89 |
| Visual Grounding | ScanRefer[52] | Failed | 3.38 |
| 3D VQA | ScanQA[53] | 26.54 | 99.89 |

Table 4: Comparison of results of Binary Locating Metric and GPT Metric of existing MLLMs. The Binary-Locating Metric is the accuracy of the predicted position, and the GPT Metric is the score from GPT response.

| | LLaVA | MiniGPT4 | mPLUG-owl | LAMM |
|---|---|---|---|---|
| Binary-Loc Metric | 14.73 | 13.12 | 4.42 | **31.2** |
| GPT Metric | **50.16** | 7.28 | 41.88 | 48.44 |

FSC147, and LSP. Since the our baseline model has been trained on a small amount of data with detection instructions, it significantly improves in localizing accuracy.

**GPT Metric.** We calculated GPT scores using a variety of tasks, including VQA, classification, captioning, as well as a small number of detection and counting tasks. As shown in Table 4, LLaVA surpasses other models in performance, while LAMM, although slightly lower than LLaVA, still outperforms Minigpt4 and mPLUG-owl by a wide margin.

### 5.4 Observation and Analysis

We conducted dozens of experiments and observations on the MLLM model across various tasks to summarize its current capabilities and limitations.

**Better Performance in Counting Tasks with Small Number of Objects.** As shown in the Table 1, recent MLLMs perform poorly on counting tasks. In the FSC147 dataset, there are data samples with dozens or even hundreds of objects, and the MLLMs would reply with "I cannot accurately count the number" for such data samples. Therefore, we conducted tests on the subset of the FSC147 dataset with less than 10 objects to evaluate the performance of the models on simple data, as shown in Figure 5 (b). The results show that the MLLMs are able to roughly estimate the number of specified objects in the image, but it is still unable to provide an exact numerical value.

**GPT Metric is More Appropriate Than BLEU.** Figure 4 illustrates the comparison between the generated captions by LLaVA and LAMM on a sample data from the Flickr30k dataset. It is evident that LAMM model produces more detailed image descriptions. However, a notable drawback is the low correlation between its generated sentences and the ground truth sentences, which consequently results in the low BLEU scores indicated in Table 1. Thus, we tried to adopt the GPT Metric to assess the relevance and accuracy of the model's output captions to the ground truth captions. GPT gives a higher score to LAMM model, compared to LLaVA, suggesting that our model is more able to generate high-quality, image-relevant text outputs. This observation also raises the possibility that using GPT-based metrics for evaluating captioning tasks instead of BLEU might offer a more effective evaluation criterion.

**Capable of Object Localization but Struggles with Precise Bounding Box Prediction.** We visualize the results of LLaVA on VOC2012 dataset. Figure 4 (a) shows that the LAMM model was able to roughly point out the bird in the image, but was unable to accurately locate the entire object.

**LAMM Model Exhibits Fine-Grained Classification Ability on CIFAR10.** As shown in Figure 4, when presented with a 32x32 pixel image of a car, the model's prediction was a more granular category: "Fiat 500L 2012", which accurately identifies the car's brand and model. The left sub figure in Figure 4 (b) shows the image of Fiat 500L 2012 on Autoevolution [54], revealing that it has very similar features to the input image from CIFAR10. These results demonstrate that the MLLM trained with our dataset has the ability to perform more fine-grained classification, and is capable of recognizing subtle differences in images and assigning them to more specific categories.

**Instruction and Reasoning Enhance Performance on SQAimage Data** Following LLaVA [15], we conducted experiments on the SQAimage dataset using different inference approaches, including prompts with or without reasoning or instruction. The prompts with reasoning make the MLLM

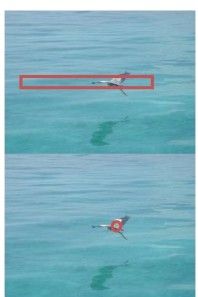

Instruction: Give the bounding box of all the objects in the image.

Response: A bird flying over the water with its wings spread out. The bird appears to be a seagull, with a white body and black wings. To provide the precise coordinates, the bird's bounding box would be (0.156, 0.477, 0.767, 0.523).

Instruction: Give the position of all the objects in the image.

Response: In the image, there is a bird flying over the water. The bird is flying low to the water, with its beak open as if it is in the process of catching a fish. The bird's position is approximately [0.55, 0.53].

(a)

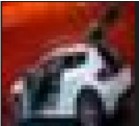
CIFAR10

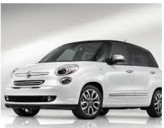
Autoevolution

Instruction: What is the category for this image?

*Ground Truth: car*

Response: Fiat 500L 2012

(b)

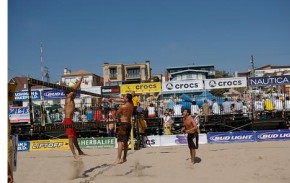

Instruction: What is the caption of this image?

*Ground Truth:*
*1. A male volleyball player in red trunks tries to block the shot of the opposing player during a professional volleyball game.*
*2. Men playing volleyball, with one player missing the ball but hands still in the air.*
*3. A group of spectators watch a men's sand volleyball game.*
*4. Volleyball players compete on a sunny day at the beach.*
*5. Men playing volleyball in the sand.*

LLaVA's Response:
The image shows a group of people playing a game of volleyball on a beach. They are actively engaged in the game, with one man jumping in the air to hit the ball. The scene is lively and energetic, capturing the spirit of outdoor sports and beach activities.

LAMM's Response:
The image captures an engaging scene from a professional sand volleyball game. Two men can be seen in the center of the court, fully immersed in the game. Both players are dressed in casual sportswear, with one wearing a white shirt and the other in a shirt of the same color. The sand court stretches across the image, providing ample space for the players to move around. There are multiple flags scattered across the scene, likely serving as markers or boundaries for the game. In the background, the sky provides a clear, light-colored backdrop to the lively action on the court.

(c)

Figure 4: Observation and analysis on various tasks. (a) Visualization results on VOC2012. (b) Visualization results on CIFAR10. The right subfigure is from [54]. (c) Results on Flickr30k.

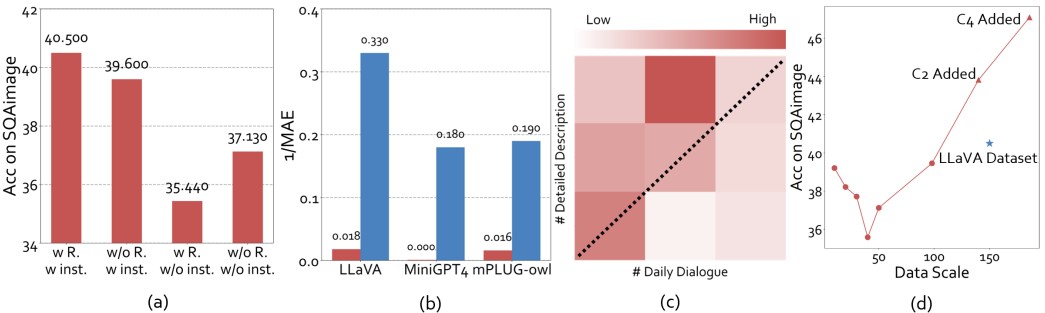

Figure 5: (a) Zero-shot Accuracy of LLaVA with different inputs on SQAimage. R. indicates reasoning and inst. indicates instruction. (b) Counting Performance on FSC147 of MLLMs. (c) Zero-shot accuracy of LAMM model trained on various data combinations on SQAimage. (d) Zero-shot accuracy of LAMM model trained additional instruction data in our dataset.

output the reasoning process before presenting the final results. The prompts with instruction give MLLM the task definition and output structure to the question to help the model better understand the task. The results in Figure 5 (a) shows that the instruction and reasoning both improve the MLLM's VQA ability. These results highlight the importance of incorporating task-specific information and reasoning process into MLLMs.

**Difficulty in Comprehending Visual Information for Domain Shifted Data.** We conducted an analysis on several datasets that exhibit significant deviations from the training dataset, including UCMerced, CelebA, and LSP. The UCMerced dataset consists of top-down views of scenes, CelebA is a facial dataset that can describe the expressions and hair colors, and the LSP dataset involves 14 key points of the human body, they are significantly different from the COCO dataset during the training phase. These results suggest that the performance of the MLLM model may degrade significantly on datasets that exhibit significant deviations from the training dataset.

**Difficulty in Reading Text on SVT data.** We analyzed the performance of our baseline model on the SVT dataset and observed unsatisfactory results in Table 1. A possible explanation is that we used the

TextVQA [55] dataset to generate visual task dialogue, which is more geared towards conversational text rather than OCR-related vision tasks. This mismatch in dataset characteristics may have resulted in suboptimal generalization of our model to the SVT dataset. To address this issue, we intend to conduct further investigations and incorporate more appropriate OCR data during the training process to improve our model's performance on OCR-related vision tasks.

**Data volume validation on SQAimage data.** As shown in Figure 5 (c) (d), our four types of image instruction tuning datasets outperform LLaVA[15] on all subsets, resulting in a 7% overall performance improvement for the complete dataset. Furthermore, we investigated the impact of sampling *Daily Dialogue* and *Detailed Description* data at different proportions. Notably, even with the small size of 10k examples, our dataset achieved comparable results to LLaVA-Dataset. As the dataset size increased, the overall performance of our model continuously improved, indicating that our dataset is scalable and can be further optimized by adding more data.

## 6 Limitations

In this part, we discuss limitation and social impact of this work from perspectives of dataset, benchmark and framework.

**Dataset** In our study, we utilized GPT-API, a state-of-the-art language model, to generate the multi-modal instruction data. To achieve the desired format, which includes multi-round dialogue and one-round detailed descriptions, we provided system messages and example dialogues as guidance for the data generation process using GPT-API. The use of GPT-API for generating text-based conversations has been widely adopted in Natural Language Processing, and previous work in multi-modal data [8, 15, 16] has demonstrated promising results in various tasks.

However, it is important to acknowledge the limitations inherent to the underlying GPT model, which are not altered by the use of GPT-API. GPT-API lacks direct access to visual information and relies solely on textual context such as captions and attributes, which restricts its understanding of images and may result in missing detailed information. While GPT-API excels at generating coherent and contextually relevant responses, it can occasionally produce responses that appear plausible but are factually incorrect or lack proper context. It may also struggle with understanding complex or ambiguous queries. Moreover, the generated data used for training may inadvertently reflect inherent biases and other truthworthy issues of GPT-API. To address ethical concerns regarding data generated with GPT-API, we performed manual sampling to examine the data, ensuring that the generated data aligns with societal values, privacy, security, toxicity, and fairness requirements and expectations. In Appendix, we provide an evaluation of the data quality and showcase additional data samples. We also transparently provide the complete prompts used to invoke GPT-API, ensuring transparency throughout our work.

**Benchmark** LAMM evaluates MLLMs on formatted computer vision tasks and datasets. Due to the diversity of language models' outputs, metrics may fluctuate across experiments. Additionally, LAMM currently adopts metrics such as GPT-eval and binary localization as an initial attempt to evaluate MLLMs' performance. Further research is needed to enhance the stability of benchmark results and design more appropriate metrics, which can be a promising direction for future investigations.

**Framework** Our work establishes a simple MLLM framework to build up a baseline model for our dataset and benchmark. However, there is potential for further development and careful design of MLLMs for future work to enhance their capabilities and performance.

## 7 Conclusion

In conclusion, our work presents LAMM, an open-source endeavor in the field of multi-modal large language models. We introduce the image and point-cloud instruction tuning dataset and benchmark, aiming to establish LAMM as a thriving ecosystem for training and evaluating MLLMs. We also provide an extensible framework to facilitate the extension of MLLMs to additional modalities. Our research showcases the effectiveness of MLLMs in handling visual modalities, including images and point clouds, and highlights their potential for generalization via instruction tuning. By making our codebase, baseline model, instruction tuning dataset, and evaluation benchmark publicly available, we aim to foster an open research community for MLLMs. We believe that our work will contribute to the advancement of MLLMs and the development of general-purpose multi-model agents.

## Acknowledgement

This work is done during Zhenfei Yin, Jiong Wang, Jianjian Cao, Zhelun Shi and Dingning Liu's internship at Shanghai Artificial Intelligence Laboratory. This work is supported in part by the National Key R&D Program of China (NO. 2022ZD0160100), and National Natural Science Foundation of China (62132001).

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

# Appendix

## A  Overview

Dataset and code in LAMM has been open sourced at https://github.com/OpenLAMM/LAMM. In this Appendix, we present construction pipeline and more examples of our dataset in Sec. B. Then, Sec. C shows details of benchmark and related evaluation metrics. Sec.D presents implementation details of our framework. Training a model based on our framework takes about 20 A100 GPU hours. Finally, more examples and results are visualized in Sec. E.

## B  Dataset

The paper introduces a novel method for constructing instruction tuning data, which represents an innovative departure from traditional techniques that rely solely on daily dialogue and detailed description. Instead, our dataset leverages additional factual knowledge extracted from Wikipedia to improve the quality and diversity of the training data. In addition, we also explore the use of traditional vision task data, covering common tasks in both 2D and 3D fields, which is converted into instruction tuning data for training purposes. By combining our new data construction method with traditional vision task data, we aim to improve the accuracy and effectiveness of instruction-tuned models in various vision-related applications. Specifically, we delve into the design of 2D and 3D portion of our dataset in Section B.1 and B.2, respectively. We also outlined the manual approach for checking the quality of the generated data in Section B.3. Finally, we provide a comprehensive explanation of the license and social impact information of our dataset in Section B.4.

### B.1  Image Instruction Tuning Dataset

***C1: n-round Daily Dialogue & C3: 1-round Detailed Description.*** The first step of our approach involves incorporating more visual information, such as visual relationships and fine-grained categories as input to GPT-API, providing dense visual context to the generated responses. To construct the *C1: n-round Daily Dialogue* and *C3: 1-round Detailed Description* data, we use the COCO images [56], similar to the LLaVA [15] approach. However, we further extract object attributes and relationships from the Visual Genome dataset [57] to emphasize fine-grained and dense information in the generated responses. Specifically, Our approach leverages image scene graph information to provide a structured representation of the objects and their relationships within the image. By doing so, we generated multi-modal dialogue data that enables us to capture the relationships between objects in the image and generate more accurate and natural language instructions. Figures 6 and 7 display the messages utilized to generate daily dialogue and detailed description data in the GPT-API. Additionally, Figure 8 provides detailed examples of the generated results for both types of data.

***C2: n-round Factual Knowledge Dialogue.*** In the second step of our approach, we expand the dataset by incorporating 42K classes of knowledge graph facts from Wikipedia using the Bamboo dataset. This addition enables MLLMs to generate question-answering data related to factual knowledge, which is a valuable addition to the dataset. To generate *C2: n-round Factual Knowledge Dialogue* data, we utilize the Bamboo dataset and Wikipedia to obtain relevant information, and then use GPT-API to generate a dialogue based on the given content. Specifically, we extract the QID labels and their corresponding Wikipedia descriptions from the Bamboo dataset to generate instruction tuning data. This approach allows us to incorporate common sense knowledge into the dataset, thereby enhancing the ability of MLLMs to generate responses that draw upon a broader range of factual knowledge. The messages used to generate factual knowledge data in the GPT-API are presented in Figure 9, while Figure 10 showcases detailed examples of the factual knowledge data generated by these messages.

***C4: 1-round Visual Task Dialogue.*** In addition to the three types of data discussed earlier, we also incorporate established computer vision tasks, such as image classification, object detection, keypoint detection, OCR, and object counting, into our dataset. This enables MLLMs to handle traditional computer vision tasks and generate responses that incorporate both language and visual information. The typical computer vision dataset consists of a set of images or videos, along with their corresponding labels or annotations that represent the desired output of the computer vision task, such as the class of objects present in the image or the location of an object. However, these discrete

```
messages = [{ "You are an AI visual assistant that can analyze a single image. You receive five sentences, each
describing the same image you are observing. In addition, specific object locations within the image are given, along with
detailed coordinates. These coordinates are in the form of bounding boxes, represented as (x1, y1, x2, y2) with floating
numbers ranging from 0 to 1. These values correspond to the top left x, top left y, bottom right x, and bottom right y. It is
worth noting that Attributes and Relationships of different objects are also given.

You need to generate a n-round Daily Dialogue data between two people. The answers should be in a tone that a visual AI
assistant is seeing the image and answering the question. Ask diverse questions and give corresponding answers. Include at
least one plausible question about the image, and provide the answer in detail. To answer such questions, one should
require first understanding the visual content, then based on the background knowledge or reasoning, either explain why
the things are happening that way, or provide guides and help to user's request. Make the question challenging by not
including the visual content details in the question so that the user needs to reason about that first.

Instead of directly mentioning the bounding box coordinates, the captions and relations, utilize those data to explain the
scene using natural language. Include questions asking about the visual content of the image, including the object types,
counting the objects, object actions, object locations, relative positions between objects, etc. Always answer as if you are
directly looking at the image. "}]

for sample in fewshot_samples:
    messages.append({"role":"user", "content":sample['context'] })
    messages.append({"role":"assistant", "content":sample['response'] })
messages.append({"role":"user", "content":'\n'.join(query)})
```

Figure 6: Messages used to construct *n-round Daily Dialogue* data for image instruction tuning.

```
messages = [{ "You are an AI visual assistant that can analyze a single image. You receive five sentences, each
describing the same image you are observing. In addition, specific object locations within the image are given, along with
detailed coordinates. These coordinates are in the form of bounding boxes, represented as (x1, y1, x2, y2) with floating
numbers ranging from 0 to 1. These values correspond to the top left x, top left y, bottom right x, and bottom right y. It is
worth noting that Attributes and Relationships of different objects are also given.

You need to generate a 1-round Detail Description data. You can use the provided caption, attributes, relationships and
bounding box information to describe the scene in a detailed manner.

Instead of directly mentioning the bounding box coordinates, utilize this data to explain the scene using natural language.
Include details like object counts, position of the objects, relative position between the objects. When using the information
from the caption and coordinates, directly explain the scene, and do not mention that the information source is the caption,
relationships or the bounding box. Always answer as if you are directly looking at the image. You can include multiple
paragraphs if necessary."}]

for sample in fewshot_samples:
    messages.append({"role":"user", "content":sample['context'] })
    messages.append({"role":"assistant", "content":sample['response'] })
messages.append({"role":"user", "content":'\n'.join(query)})
```

Figure 7: Messages used to construct *1-round Detailed Description* data for image instruction tuning.

results are not suitable for large language model dialogues, as they do not allow for natural language
interactions. To address this issue, our proposed approach involves converting computer vision tasks,
such as image classification, into natural language dialogues to enable large language models to
perform these tasks through dialogue interactions. In detail, we first use GPT-API to generate a
template pool of questions and answers for each task. Then, we randomly select a pair from the
question template pool and answer template pool to combine with a piece of data from the computer
vision dataset, creating the *C4: 1-round Visual Task Dialogue* data. Figure 27-30 provide some
examples of the dialogues generated using our proposed approach for converting computer vision
tasks into natural language dialogues. This approach allows us to leverage the rich visual information
in traditional computer vision datasets and incorporate it into the instruction tuning process, thereby
enhancing the ability of MLLMs to understand and respond to natural language instructions related
to these tasks.

In summary, the construction of 2D part in our dataset provides a comprehensive and diverse samples
of real-world scenarios, incorporating fine-grained and dense information from object relationships
and factual knowledge sources. The dataset contains 186K unique language-image instruction-
following samples, including 49K in daily dialogues, 49K in detailed descriptions, 42K in factual
Knowledge dialogues, and 46K in visual task dialogues. Our experiments showed that the use of
GPT-API consistently provides higher-quality instruction-following data, such as spatial reasoning.

**Captions:** [A clock on a tower with a church steeple and bell. A clock tower has a fenced in bell on top. A very tall clock tower with a large bell sitting above it. The cross, bell and clock on the tower over a church. There is a tower clock on the wall.]
**Objects:** bell: rusty, [0.4, 0.23, 0.57, 0.33]. clock: white, black, [0.18, 0.65, 0.77, 0.89]. building: white, tall, rusty, [0.14, 0.09, 0.9, 0.99]. sky: blue, [0.01, 0.0, 1.0, 0.7].

*C1: n-round Daily Dialogue*
Instruction: <image> What kind of building has a clock tower in the image?
Response: In the image, there is a church with a clock tower.
Instruction : Is there anything special about the clock tower?
Response : Yes, there is a large, fenced-in bell on top of the clock tower, which makes it unique.
Instruction : What color is the clock on the tower?
Response : The clock on the tower is white with black numerals and hands.
Instruction : Can you describe the appearance of the sky in the image?
Response : The sky in the image is blue.
Instruction : What role might this clock tower and bell play in the local community?
Response : The clock tower and bell might play a significant role in the local community by serving as a landmark and a focal point of the town or neighborhood. The clock tower, with its large, visible clock face, helps the residents keep track of time and maintain their daily schedules. In addition, the bell might be used to signal important events, such as religious services, weddings, or funerals. It could also serve as a reminder of community values and traditions, as well as a symbol of unity and shared history.

*C3: 1-round Detailed Description*
Instruction : <image> What can you observe in this image?
Response : The image presents a tall, white clock tower that stands out against a clear blue sky. The clock face is prominently visible, featuring white background and black hands. Above the clock, a fenced-in, rusty bell can be seen, adding an air of history to the tower. Furthermore, the tower also displays a cross at the top, signifying its connection to a church. The combination of the cross, bell, and clock creates a striking visual that showcases the tower's architectural and historical significance.

Figure 8: Example of generated *n-round daily dialogue* and *1-round detailed description* data.

```
messages = [{"You are an AI visual assistant that can analyze a single image. You receive a category label for an image
and its associated English description describing the same image you are looking at. The category information is the
category label of the image you are seeing. In addition, The English description is a detailed semantic information for the
given category.

You need to generate a n-round Factual Knowledge Dialogue data based on the received category lable and its English
description. Please ask questions around the given category information and its corresponding description information, and
don't ask questions that cannot be reasoned based on the given information. Questions and answers should be concise and
effective, with reasoning logic. Answer all questions as you are seeing the image.

The answers should be in a tone that a visual AI assistant is seeing the image and answering the question. Ask diverse
questions and give corresponding answers. Include questions asking about the visual content of the image. Only include
questions that have definite answers: (1) one can see the content in the image that the question asks about and can answer
confidently; (2) one can determine confidently from the image that it is not in the image. Do not ask any question that
cannot be answered confidently. Also include complex questions that are relevant to the content in the image, for example,
asking about background knowledge of the objects in the image, asking to discuss about events happening in the image, etc.
Again, do not ask about uncertain details. Provide detailed answers when answering complex questions. For example, give
detailed examples or reasoning steps to make the content more convincing and well-organized. You can include multiple
paragraphs if necessary "}]

for sample in fewshot_samples:
    messages.append({"role":"user", "content":sample['context'] })
    messages.append({"role":"assistant", "content":sample['response'] })
messages.append({"role":"user", "content":'\n'.join(query)})
```

Figure 9: Messages used to construct *n-round Factual Knowledge Dialogue* data for image instruction tuning.

These features make our dataset a valuable resource for researchers and practitioners working in the computer vision and natural language processing fields.

## B.2 Point Cloud Instruction Tuning Dataset

The construction pipeline of point cloud instruction tuning data is similar to that of image instruction tuning data. However, due to the limited availability of 3D data, point cloud instruction tuning dataset only consists of three major components: n-round plain conversation and 1-round detailed description data from GPT-API and 1-round visual dialogue data converted from datasets for existing 3D vision tasks.

***C1: n-round Daily Dialogue & C3: 1-round Detailed Description.*** To construct the *C1: n-round Daily Dialogue* and *C3: 1-round Detailed Description* data, we choose point clouds from 3RScan [59]

Figure 10: The example for constructing *n-round Factual Knowledge Dialogue* data. The description is from Wikipedia page.

Figure 11: Message to transfer visual question answering annotations from CLEVR3D [58] to declarative sentences for 3D data.

as data source and use its original 3D bounding box annotations. Since there is no caption annotation for 3RScan, we input visual question answering (VQA) annotations from CLEVR3D [58] to GPT-API and ask it to convert the Q&A data into declarative sentences, which serves as point cloud captions in further steps. Figure 11 shows the corresponding prompts. Object attributes and relationships are extracted from scene graph annotation in 3DSSG [60]. Figure 12 and 13 show the prompts to let GPT-API generate daily dialogue and detailed description data for point clouds. Since full annotation of a scene point cloud may easily exceed input token limits of GPT-API, we randomly selected 10 captions and keep bounding box and relationships of corresponding objects as input contexts. For GPT-generated data, We limit the number of turns in each dialogue data to no more than 10, and any data exceeding this limit will be split into different samples. Figure 14 shows an example of GPT-generated data.

***C4: 1-round Visual Task Dialogue.*** On the other hand, we also leverage annotations for existing 3D vision tasks, such as point cloud classification, 3D object detection, and CLEVR3D for 3D VQA. Similar to 2D datasets, we designed 15 templates for instruction and response by sending definitions of the corresponding tasks to GPT-API. Then instruction data are formulated by replacing keywords with corresponding annotations. Templates of 3 tasks involved are presented in Figure 31, 32 and 33, respectively.

```
messages = [{"You are an AI visual assistant that can analyze a single point cloud. You receive five sentences, each describing the same
point cloud you are observing. In addition, specific object locations within the point cloud are given, along with detailed coordinates.
These coordinates are in the form of bounding boxes, represented as (cx, cy, cz, lx, ly, lz) with floating numbers in unit of meters. These
values correspond to the top x, y, z coordinates of bounding box ceter and length of bounding box along x, y, z axis. It is worth noting
that Attributes and Relationships of different objects are also given.
Generate some questions and answers of the scene in a detailed manner. Instead of directly mentioning the bounding box
coordinates or captions given, you should utilize this data to explain the question using natural language. Include details like the
scenario, object counts, position of the objects, relative position between the objects. When using the information from the caption and
coordinates, directly explain the scene, and do not mention that the information source is the caption, relationships or the bounding
box. Answer questions or descriptions as if you really saw the whole scene, using the tone of seeing the scene to ask questions or
answer. Please ask questions around the given category information and its corresponding description information, and don't ask
questions that cannot be reasoned based on the given information. Questions and answers should be concise and effective, with
reasoning logic. Answer all questions as you are seeing the point cloud. The answers should be in a tone that a visual AI assistant is
seeing the point cloud and answering the question.
Ask diverse questions and give corresponding answers. Include questions asking about the visual content of the point cloud.
Only include questions that have definite answers: one can see the content in the point cloud that the question asks about and can
answer confidently. All descriptions are attributes or relationships with other objects, which does not mean that there are people in the
scene."}]
for sample in fewshot_samples:
    messages.append({"role":"user", "content":sample['context']})
    messages.append({"role":"assistant", "content":sample['response']})
messages.append({"role":"user", "content":'\n'.join(query)})
```

Figure 12: Message to generate *n-round Daily Conversation Dialogue* data in 3D portion of our dataset.

```
messages = [{"You are an AI visual assistant that can analyze a single point cloud. You receive five sentences, each
describing the same point cloud you are observing. In addition, specific object locations within the point cloud are given,
along with detailed coordinates. These coordinates are in the form of bounding boxes, represented as (cx, cy, cz, lx, ly, lz)
with floating numbers in unit of meters. These values correspond to the top x, y, z coordinates of bounding box ceter and
length of bounding box along x, y, z axis. It is worth noting that Attributes and Relationships of different objects are also
given.
You need use the provided caption, relationships and bounding box information to describe the scene in a detailed
manner. Instead of directly mentioning the bounding box coordinates or captions given, you should utilize this data to
explain the scene using natural language. Include details like the scenario, object
counts, position of the objects, relative position between the objects. When using the information from the caption and
coordinates, directly explain the scene, and do not mention
that the information source is the caption, relationships or the bounding box. Answer questions or descriptions as if you
really saw the whole scene, using the tone of seeing the scene to ask questions or answer. Always answer as if you are
directly looking at the point cloud. You can include multiple paragraphs if
necessary. Every question and answer must be related."}]
for sample in fewshot_samples:
    messages.append({"role":"user", "content":sample['context']})
    messages.append({"role":"assistant", "content":sample['response']})
messages.append({"role":"user", "content":'\n'.join(query)})
```

Figure 13: Message to generate *1-round Detailed Description* data in 3D portion of our dataset.

Finally, 3D portion of our dataset contains 10K samples in total, and the number of ShapeNet, 3RScan detection, CLEVR3D, and GPT-generated dialogue are 2K, 1.3K, 2K, and 4.9K, respectively.

## B.3 Quality Check

In order to ensure the quality of the generated instruction tuning data, we implemented several measures. Firstly, we generate a small amount of data as a cold start and conduct manual check on the generated data. This involved carefully assessing the quality and making necessary adjustments to the message information provided as input to GPT-API. The iterative process aimed to eliminate ethical concerns and establish a strong correlation between the generated data and the corresponding inputs. We repeated this process until the desired level of quality was achieved. Once satisfied, we proceeded to generate a large volume of data. Furthermore, to verify the quality of the generated dataset, we randomly select a subset of 10% data for manual checks. This step allowed us to evaluate the generated data against our specific requirements and quality standards. During this evaluation, any formatting issues or incorrect answers generated by GPT-4 were filtered out to ensure the usability and reliability of the data. By combining manual checks during the iterative generation process and

Figure 14: Example of GPT-generated n-round daily dialogue and 1-round detailed description data in 3D portion of our dataset.

subsequent random manual checks on the final dataset, we strive to ensure that the generated data meets our rigorous quality standards and aligns with the specific needs of our dataset.

### B.4   Social Impact

Our dataset is a compilation of publicly available datasets that have been licensed under the Creative Commons license (CC-BY). We have taken great care to follow all necessary legal protocols to use this data in our research, and believe that transparency in data licensing is crucial for ensuring proper attribution and appropriate use of the data. Besides, the dataset includes images sourced from publicly available datasets and language data generated using the GPT-API. While we have taken steps to ensure appropriate content, we acknowledge that problematic content may exist. If you encounter any such content, please notify us immediately, and we will make necessary modifications to maintain a high-quality dataset that is free of inappropriate content. To protect the privacy of individuals and vehicles captured in the images, we plan to obfuscate sensitive information, such as faces and license plates, before publishing the dataset. We are committed to maintaining a dataset that is both high-quality and ethically responsible and pledge to uphold principles of privacy and transparency in our work.

## C   Benchmark

### C.1   Benchmark on image tasks

We selected a set of nine commonly used CV tasks to evaluate the performance of MLLM models in our benchmark on image tasks. Our task selection criteria were based on widely studied tasks in the CV field that can showcase the MLLM model's abilities in visual interpretation, localization, and question-answering. Table 5 provides a summary of the tasks and the corresponding common evaluation metrics, which are based on the output that the MLLM models are required to generate for each task. We utilized a prompt-based approach to instruct the MLLM models to understand the task definition and generate the desired output. The ability of the models to understand and interpret the given instruction was also evaluated as part of the assessment criteria. As the models' outputs are text, we use different text-processing techniques for each task to extract entities as the final answers for evaluation. For each task, We selected datasets that are distinct from the training datasets, as our benchmark evaluation is conducted in an out-of-distribution zero-shot setting.

Table 5: CV tasks in Our Benchmark

| Task | Output | Metrics |
|------|--------|---------|
| Classification | label name | Acc |
| Detection | list of object label and bbox | mAP50 |
| VQA | option and answer | Acc |
| Image Caption | captions | BLEU4 |
| Fine-grained classification | fine-grained label name | Acc |
| Object counting | number | MAE |
| OCR | list of words | word Acc |
| Facial classification | answer | Acc |
| Keypoints detection | keypoints | PCK |
| 3D Detection | list of object label and bbox | mAP50 |
| 3D VQA | option and answer | Acc |
| 3D Visual Grounding | bbox | mAP50 |

**Classification** This task involves predicting the most likely category label for an image. For MLLM models, the task involves performing open-vocabulary classification. We selected CIFAR-10 [41] as the test dataset for the evaluation of classification. CIFAR10 contains 10000 test images across 10 common categories. We utilize NLTK to extract noun entities from the models' output text, and expand them to a synonym set for accuracy evaluation calculation.

**Object Detection** We selected the VOC 2012[42] datasets to evaluate the model's ability to detect objects in images while considering both its visual interpretation and localization capabilities. To evaluate the accuracy of object category predictions, we employ a similar approach to classification tasks. We also use regular expression matching to extract the models' output bounding boxes for mAP50 calculation.

**Visual Question Answering** We selected the ScienceQA[43] and AI2D[44] datasets to evaluate the MLLM model's ability to answer questions about images. The ScienceQA and AI2D datasets include over 2017 and 5793 multiple-choice questions with images, respectively. We extract the image-containing data from the ScienceQA dataset to create the SQAimage dataset. We then tested MLLM models on the SQAimage dataset to evaluate their multimodal understanding skills. As both ScienceQA and AI2D datasets are presented in a multiple-choice format, we evaluated the model's performance using the accuracy metric. Following LLaVA [15], we prompt the MLLM to output the complex reasoning procession, followed by the final option answer.

**Image Caption** The image caption task involves generating a textual description of an image. We selected the Flickr30k[45] dataset to evaluate the MLLM model's ability to understand images and generate descriptive captions. Flickr30k contains a variety of objects and scenes with diverse captions, providing a challenging task for the MLLM model. To evaluate the quality of the models' text outputs, we split the generated text into sentences and calculate the BLEU-4 score for each. The highest score is selected as the final result.

**Fine-grained classification** Similar to the classification task, the fine-grained classification task requires the model to make predictions across a large number of fine-grained categories. We selected UCMerced Land Use dataset [46] as the test set. UCMerced Land Use contains 21 classes of land-use categories, including airports, forests, and residential areas. Similar to classification, we report Accuracy.

**Object counting** We selected the FSC147 dataset for object counting evaluation. FSC147[47] is a dataset of 1190 images containing various objects, including animals, vehicles, and household items. The images in this dataset are challenging and contain occlusions and overlapping objects, making it a suitable choice to test the model's object recognition and localization capabilities. We utilize regular expression matching to extract the numeric entity and evaluate the model's performance using the mean absolute error (MAE) metric.

**Optical Character Recognition** The OCR (Optical Character Recognition) task involves recognizing and transcribing text from images. To evaluate the MLLM model's ability to recognize text from images, we selected SVT dataset [48]. We extract the entities enclosed in quotation marks from the generated text as the predicted word list. Word Accuracy is adopted as the evaluation metric.

```
[System Message]
### Human: <vision> [Vision Tokens] </vision> [Instructions]
### Assistant: [Response a]
### ANSWER: [Response b]
```

Figure 15: Template instructions for VQA inference. "Response a" is the generated reasoning process, which is the output of the first inference. "Response b" is the output answer, which is the ouput following the prompt "### ANSWER".

**Facial Classification** Due to the difficulty of performing face recognition tasks using MLLM, we evaluated the model's performance on facial attribute classification tasks. We selected the CelebA[49] dataset, which contains 19962 images for testing with annotations for 40 facial attributes, including hair color and facial expression. Specifically, we evaluated the model's ability to predict whether a person in an image is smiling, named CelebA(Smile) dataset, and the color of their hair, named CelebA(Hair) dataset. We aimed to evaluate the MLLM model's ability to understand facial images. Classification accuracy is used as the evaluation metric.

**Keypoints Detection** To evaluate the models' ability to perform fine-grained point localization, we utilized the LSP[50] dataset for keypoint detection. To simplify the task difficulty for MLLM models, we employed a grounding approach, where we sequentially asked the model to predict the position of each human body keypoints in the image. The evaluation metric used for this task was PCK (Percentage of Correct Keypoints).

### C.2    Inference Details

### C.2.1    System messages for image tasks

Figure 17 shows the system messages defined for each image task. The system messages, which include the task definition and the output structure, is a part of the instruction that prompt the MLLM models to generated responses. This is designed to enable the model to better understand the task it is performing, focus on the critical aspects, and output the appropriate structure. Note that some tasks do not require a defined output structure. In such cases, the model can output any text as a response.

### C.2.2    Instructions for VQA

Different from other common image tasks, besides the system messages designed in C.2.1, we prompt MLLM to generate the reasoning process additionally, as figure 15 shows. To prompt the model to output its reasoning process, we first use conventional instruction texts to generate "Response a". We then combine the first instructions , the "Response a", and the prompt "### ANSWER" to make the model generate the option as the final answer.

### C.2.3    Metrics

Our benchmark includes two evaluation settings. The first is a zero-shot setting, where we selected downstream tasks that have no intersection with the MLLM's training data. We provide the zero-shot results of the current MLLM models on these datasets. The second setting involves fine-tuning on mainstream task datasets, covering tasks such as detection, classification, and VQA.

### C.2.4    Binary Locating Metric

The ability to accurately localize objects in an image is a crucial component of MLLM models' visual understanding skills. In addition to using conventional detection tasks to calculate mAP, we attempted a more direct method for evaluating the models' localization ability, namely Binary Locating Metric. Distinct from object detection, which requires the model to output a bounding box, we instructed the model with "output the position of the object" instead of "output the bounding box of the object" to output the approximate position. During the evaluation phase, the model's predicted keypoint was considered correct as long as it was within the object's bounding box. Object locating is evaluated on all datasets involving object localization, including object detection, object counting, and keypoints detection. Compared to the traditional detection evaluation methods, the object locating evaluation

method provides a more reasonable and direct approach for evaluating the localization ability of MLLM.

### C.2.5 GPT Metric

To evaluate the overall understanding and question-answering abilities of MLLM models, we utilized the GPT Metric. Unlike LLaVA[15] and Vicuna [9], we ranked the answers of multiple models using GPT. Similar to the pipeline approach, we give GPT an instruction, informing it of the task definition, the question, and the answer provided by each model. We then ranked each model's response based on its relevance and accuracy with the answer. Each model received a score based on its ranking, and the average score obtained on all test data served as a metric for measuring the model's overall ability. Our GPT evaluation datasets cover various visual tasks, including captioning and VQA tasks involving image description and answering, as well as a small number of detection and counting tasks related to object localization.

### C.3 Benchmark on point cloud tasks

For benchmark on point cloud tasks, we focus on three tasks of scene perception, including 3D object detection, visual grounding, and 3D visual question answering. Figure 18 presents system messages for point cloud tasks.

**3D Object Detection**. As it's widely used in 3D object detection, we select ScanNetv2 [51] as the dataset to evaluate MLLM's ability to locate objects in a point cloud and identify semantics, whose validation set contains 312 scenes. In this task, MLLM is expected to list all objects along with bounding boxes, and we extract bounding boxes from the response text by entity extraction. Boxes whose IoU with ground truth is larger than 50% count for positive predictions and we use mean Average Precision (mAP) to evaluate performance.

**Visual Grounding**. This task aims to locate the object described by a given caption and output the corresponding bounding box. We test on ScanRefer [52] in this task, which provides human-labeled captions towards each object in ScanNet and its test set contains 9508 samples. Similar with object detection, mean average precision (mAP) is reported to evaluate MLLM's capacity.

**3D Visual Question Answering**. ScanQA [53] is proposed for 3D visual question answering before, and models are required to answer the given questions based on the point cloud. It has been formatted as an attribute classification task in previous work [53]. However, MLLM's output cannot be constrained with several classes consistently and is usually long text to explain details, so the original metrics in ScanQA, Exact Matching & BLEU, cannot be used for test, as long text is different from the style of given ground truth and the BLEU score inevitably decreases for long-text results. Following ScienceQA in 2D VQA task, we transfer this task to be a multiple-choice problem. First, we feed the original question-answer pairs to GPT-API and ask for 5 confusing options. Then MLLM is expected to choose the correct option or output the correct content. Thus, a metric of accuracy is used to evaluate model performance.

**Evaluation Settings** Similar to evaluation for 2D tasks, our 3D benchmark includes two settings for evaluation. The first one is a zero-shot setting. MLLM is trained on instruction data from 3D portion of our dataset, whose point clouds come from 3RScan or ShapeNet and has no overlap with ones in downstream tasks. Furthermore, we finetune the models trained on our 3D datasets by training a set of downstream tasks and reporting metrics on the corresponding test set.

## D  Implementation Details

In our experiments, 2D and 3D models are trained independently, and only the feature projection layer and LoRA parameters are optimized during training while LLM can be shared among tasks.

For all experiments, trainable parameters are optimized by Adam optimizer with a learning rate initialized to be 5e-4, and scheduled using a linear decay scheduler. We For 2D experiments, models are trained for 2 epochs. For 3D experiments, we increase the number of iterations to 10,000 in case of too few samples. We use 4 A100-80GB to conduct experiments. Each GPU process 2 samples every iteration and the effective batch size are set to 64 by gradient accumulation. For reference, 2D

```
[System Message]
### Human: <vision> [Vision Tokens] </vision> [Instructions]
### Assistant: [Response]
```

Figure 16: Template for multi-modal data pairs. **Bold words** stand for corresponding text data and italic words indicate fixed templates. $< vision >$ & $< /vision >$ stand for start & end token for vision contents.

experiments at most last for about 8 hours for 186K samples, while 3D experiments require about 3 hours.

Following Vicuna [9], we format multi-modal training data as Figure 16. $[SystemMesssage]$ specifies the corresponding task of sample, $[Query]$ refers to position of texts from human and $[Response]$ refers to contents expected for LLM. The special tokens $< vision >$ & $< /vision >$ represents start and end positions for vision content. We use $< Img >< /Img >$ and $< Pcl >< /Pcl >$ in 2D and 3D datasets, respectively. The training objective used is next token prediction loss, and only text tokens of $[Response]$ count for loss computation. As we use CLIP [38] pre-trained ViT-Large-14 as visual encoder, the number of vision tokens are 256 and length of text tokens after vision tokens are limited to $400$ in training.

# E    Demonstrations

## E.1    Results on CIFAR10

Figure 19 presents some examples responses from model trained by our dataset on CIFAR10, where the model's answers were judged as incorrect in the evaluation, but in fact, our model provided a more granular classification result. The left column shows the test images from CIFAR10, and the right column displays the images of the objects that the model classified, including toad [61], Land Rover Series II [62], Mirage 2000D fighter aircraft [63] and police car [64]. It is evident that the fine-grained objects classified by our model have very similar features to the input images, demonstrating its ability to perform fine-grained classification.

## E.2    More detailed information on image caption

Our model performed poorly on the Flickr30k dataset in terms of BLEU scores. This is because model's responses include additional details that are not captured by the ground truth captions. Figure 20 illustrates this phenomenon, where the highlighted text in red represents the matching ground truth captions, while the text in orange is not matched but is still relevant to the image content. It is evident that our model is capable of providing more detailed descriptions of the image, which is not captured by the traditional BLEU metric.

## E.3    Comparison with LLaVA on detection and counting tasks

We compared the performance of model trained by our dataset with that by LLaVA on both object detection and counting tasks. Figure 21 illustrates the comparison results on detection, where the leftmost images represent the ground truth bounding box, and the rightmost images show the visualizations of the responses after entity extraction.

Although LLaVA was able to identify the approximate location of the object, it was unable to provide precise bounding box coordinates. On the other hand, our model demonstrated superior detection capabilities after fine-tuning on detection-related data and was able to provide more accurate bounding box coordinates. Additionally, our model also exhibited better counting performance, as shown in Figure 22. It is worth noting that counting is essentially a task that tests the model's localization ability.

## E.4    Results of binary-loc metric and GPT metric

We present the results of our model and LLaVA on the binary locating metric in Figure 24 (a), where our model demonstrates more precise localization abilities. The green points in the image are

the visualization of the predicted key points. In the second row of the figure, our model outputs a bounding box, which we break down into two position coordinates (top-left and bottom-right) during entity extraction.

In Figure 24 (b), we show the evaluation results of the two models' image captioning responses using the GPT metric. The GPT metric considers our model's responses to be more specific and accurate compared to LLaVA, resulting in a higher ranking. These results further demonstrate the effectiveness of the model trained on our dataset in accurately detecting, locating, and describing objects in images.

### E.5 More demonstration examples

Figure 23 shows the results of our model on VQA task and Figure 25 shows its example results on 3DVQA task. Figure 26 shows the results on in-the-wild images.

## Classification

Your primary objective as an AI assistant is to perform a classification task accurately and reliably, as this information is crucial for users to make informed decisions based on image data. To simply providing a class label for a given image, ensure that the classification is dependable and precise.

Please provide a label that accurately describes the subject of the image.

## Detection

You are now performing an object detection task, and your goal is to locate all instances of objects in an image, such as people, cars, animals, or other objects, and give the corresponding coordinates.

These coordinates are in the form of bounding boxes, represented as (x1, y1, x2, y2) with floating numbers ranging from 0 to 1. These values correspond to the top left x, top left y, bottom right x, and bottom right y.

## VQA

As an AI assistant, you are performing a visual question answering task, and your goal is to generate natural language answers that accurately solve the question. In order to generate accurate answers to questions about visual content, you must be able to understand the content of images, understand the meaning of questions, and perform complex reasoning processes.

## Image Caption

As an AI assistant, your primary task is to perform image captioning, which requires you to generate clear and concise natural language descriptions of the visual content. To achieve this, you must be able to understand the visual content of the image, identify its salient features, and generate a coherent and contextually relevant caption that accurately conveys its meaning.

Generate descriptions of the visual context.

## Fine-grained Classification

As an AI assistant, your primary task is to perform image captioning, which requires you to generate clear and concise natural language descriptions of the visual content. To achieve this, you must be able to understand the visual content of the image, identify its salient features, and generate a coherent and contextually relevant caption that accurately conveys its meaning.

Please provide a fine-grained label that accurately describes the subject of the image.

## Object Counting

As an AI assistant, you are performing an object counting task. Your goal is to accurately count the number of objects in an image. Object counting is a computer vision task that involves detecting and counting the number of instances of specific objects within an image. You need to analyze the input image and accurately count the number of objects in it.

Give me a precise numerical result.

## OCR

You are performing an Optical Character Recognition task, which involves recognizing and extracting text from images. To generate accurate answers to questions about the text content of images, you must be able to accurately recognize and extract text from images, and understand the meaning of questions.

Your answer must be a list of words.

## Facial Classification

You are performing an Optical Character Recognition task, which involves recognizing and extracting text from images. To generate accurate answers to questions about the text content of images, you must be able to accurately recognize and extract text from images, and understand the meaning of questions.

## Keypoints Detection

You are an AI visual assistant that can analyze a single image and detect human key points. You will be provided with an image and specified which human body parts the user want you to detect. To generate accurate answers, you must be able to understand the content of images and comprehend the meaning of questions.

Please express the location as [x, y]. Both x and y are ratios between 0 and 1, indicating the position relative to the entire image.

Figure 17: System messages for benchmark on image tasks

### 3D object Detection

You are now performing an object detection task, and your goal is to locate all instances of objects in a point cloud, such as furniture, transportations, or other objects, and give the corresponding coordinates.

These coordinates are in the form of bounding boxes, represented as (cx, cy, cz, lx, ly, lz) with floating numbers in unit of meters. These values correspond to the x, y, z coordinates of bounding box center and length of bounding box along x, y, z axis.

### 3D Visual Grounding

You are able to understand the visual content that the user provides, and assist the user with a variety of tasks using natural language. You are now performing an visual grounding task, and your goal is to locate the instances of objects I want with the given description in a point cloud, such as furniture, or other objects, and give the corresponding coordinates.

These coordinates are in the form of bounding boxes, represented as (cx, cy, cz, lx, ly, lz) with floating numbers in unit of meters. These values correspond to x,y,z coordinates of center of bounding box and length of bounding box along x, y, z axis

### 3D VQA

You are able to understand the visual content that the user provides, and assist the user with a variety of tasks using natural language. You are now performing an visual question and answering task, and your goal is to generate natural language answers that accurately solve the question. In order to generate accurate answers to questions about visual content, you must be able to understand the content of point cloud, understand the meaning of questions, and perform complex reasoning processes.

Figure 18: System messages for benchmark on point cloud tasks

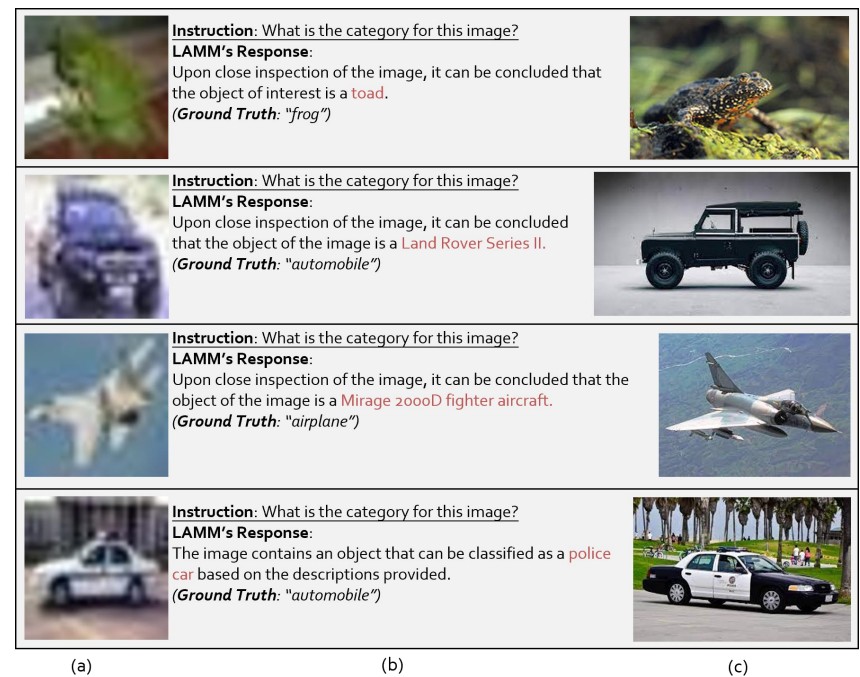

(a)                                        (b)                                        (c)

Figure 19: Results of model trained by our dataset on CIFAR10. (a) The images from CIFAR10 test set. (b) The instruction, response from our model and the ground truth. (c) The reference images.

**Instruction**: What is the caption of this image?

*(**Ground Truth**:*

*1. A man standing in a bar with a neon beer sign hanging on the wall.*
*2. A man stands next to three video game machines and a beer sign.*
*3. A man stands next to a bank of computer gambling machines.*
*4. A man is standing by a group of video games in a bar.*
*5. A man stands next to three video machines. )*

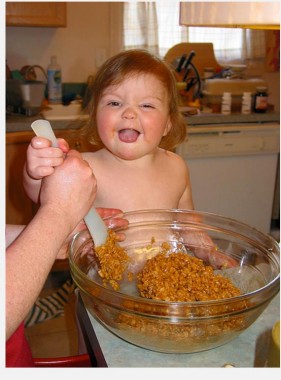

**LAMM's Response:**
The caption of this image is "This vibrant image features a man standing next to a slot machine in a casino. He appears to be checking his smartphone while wearing glasses. The slot machine is part of a row and the man is standing to the left of it. The background of the image is predominantly blue, with a white ceiling and lights installed on the machine. The slot machine itself has various buttons and symbols, including a circle, the number 7, and an orange circle. There is also a sign above the machine."

**Instruction**: What is the caption of this image?

*(**Ground Truth**:*

*1. A toddler grasping the handle of a tool to stir food in a bowl along with*
*another person whose arms are seen holding the bowl and tool.*
*2. A naked baby makes a face while helping someone stir food.*
*3. A toddler is cooking with another person.*
*4. Toddler is helping to stir ingredients.*
*5. A toddler mixes some food in a bowel.)*

**LAMM's Response:**
The image depicts a young girl in a kitchen, seemingly learning about cooking. She is standing at a counter in front of a bowl of food. The bowl is filled with a variety of ingredients, including chopped onions and a dark purple fruit. The girl is holding a spatula, likely engaging in some form of cooking activity. The background reveals a sink, further suggesting the kitchen setting.

**Instruction**: What is the caption of this image?

*(**Ground Truth**:*

*1. A man sits on a bench holding his dog and looking at the water.*
*2. A man holding a dog sitting on a bench overlooking a lake.*
*3. A man and a dog sit on a bench near a body of water.*
*4. A man and his dog watch the sunset from a bench.*
*5. A man holds his dog near a pond. )*

**LAMM's Response:**
This image captures a heartwarming scene of a man, seemingly lost in thought, holding a small black and white dog on his lap. The man and the dog are seated on a brown wooden bench, which occupies most of the lower portion of the frame. The bench is situated against a backdrop of grass, with a tree located slightly to the right of the bench. The image exudes a sense of tranquility and companionship between the man and his dog.

Figure 20: Our model's Response on fickr30k dataset. The highlighted text in red represents the matching ground truth captions in BLEU evaluation. The text in orange is not matched but is still relevant to the image content.

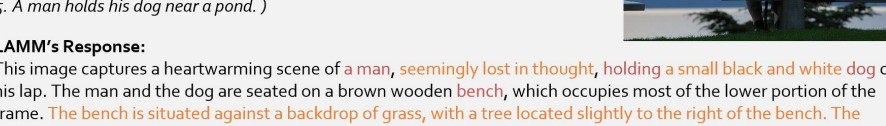
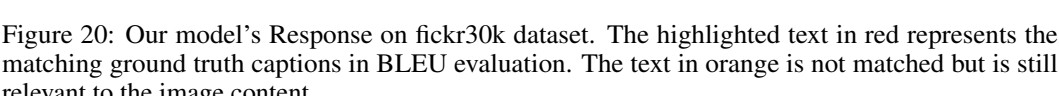

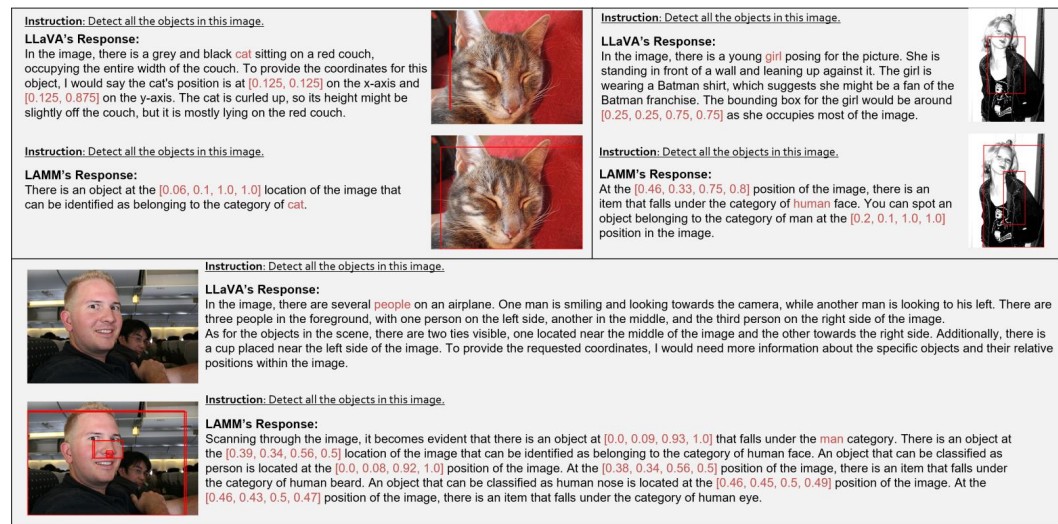

Figure 21: Comparison of models trained on our dataset and LLaVA on VOC2012.

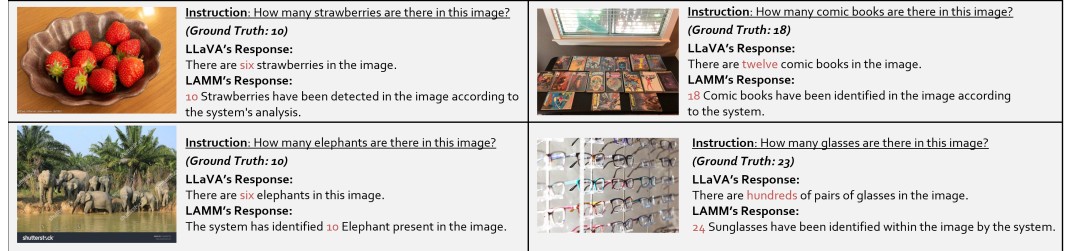

Figure 22: Comparison of models trained on our dataset and LLaVA on FSC147.

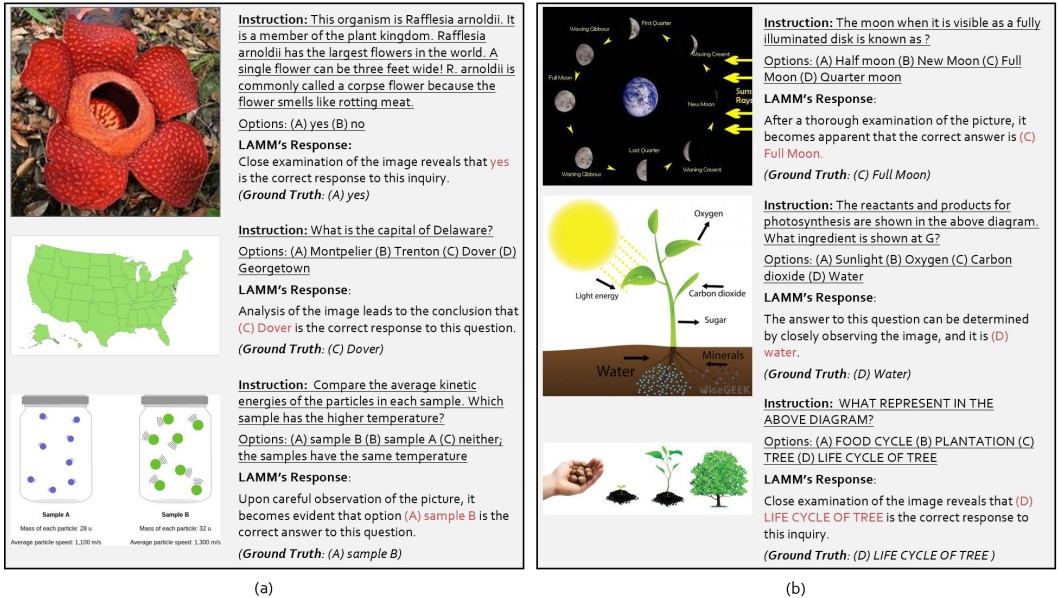

(a)                                                                    (b)

Figure 23: (a) Example results of models trained on our dataset on SQAimage. (b) Example results of our model on AI2D.

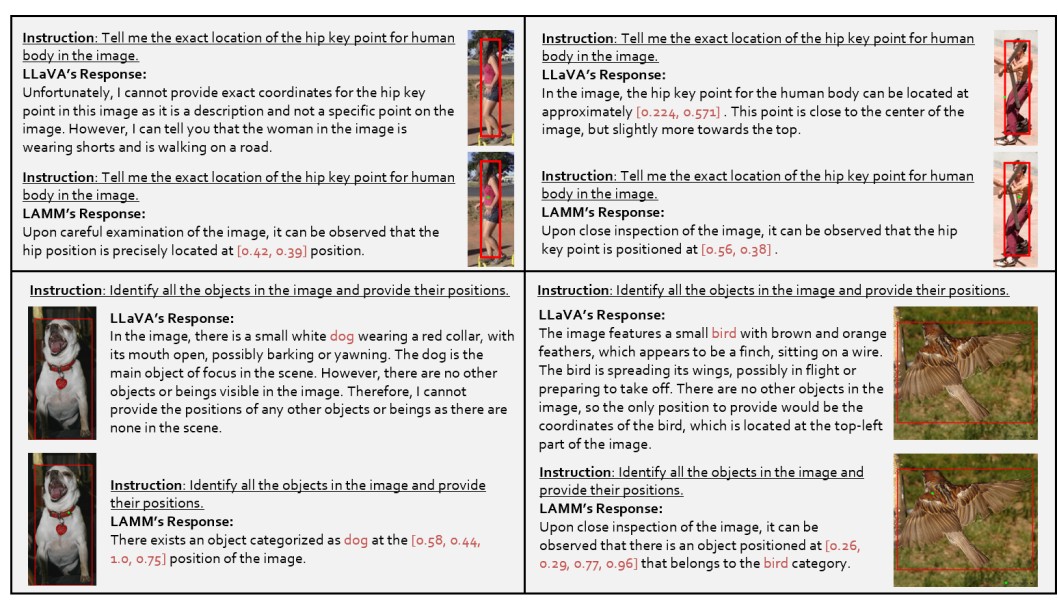

(a)

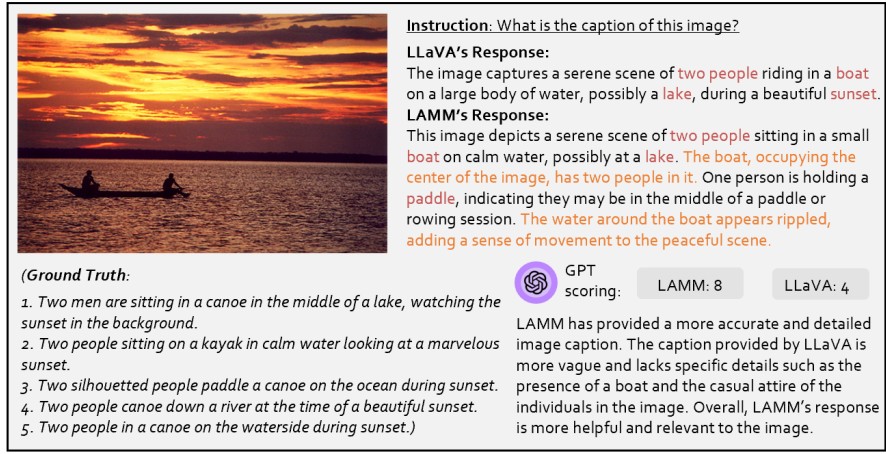

(b)

Figure 24: Comparison of models trained on our dataset and LLaVA on binary-loc metric and GPT metric. (a) The comparison on binary-loc metric. (b) The results of GPT metric.

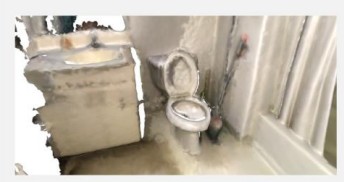

**Instruction:** What shape does the toilet have?
Options: (A) oval shape toilet. (B) a star shape. (C) a heart shape. (D) a triangular shape.(E) a diamond shape (F) a rectangular shape
**LAMM's Response:**
Upon careful examination of the point cloud, it becomes evident that the answer to this question is (A) oval shape toilet.

(*Ground Truth*: (A) oval shape toilet)

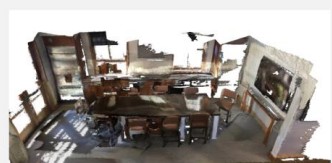

**Instruction:** What is on the wall of the kitchen?
Options: (A) a window. (B) a bookshelf. (C) a clock. (D) tv. (E) a family photo. (F) a painting.
**LAMM's Response:**
By closely observing the point cloud, one can derive that the answer to this question is (D) tv.

(*Ground Truth*: (D) tv)

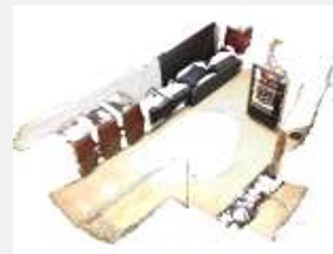

**Instruction:** What objects are sitting on the black couch?
Options: (A) dining table. (B) blue curtains. (C) ceiling fan. (D) 2 black cushions. (E) bicycle. (F) TV remote
**LAMM's Response:**
By carefully scrutinizing the details in the point cloud, it can be concluded that (D) 2 black cushions is the answer to this question.

(*Ground Truth*: (D) 2 black cushions)

Figure 25: Example results of our model on ScanQA.

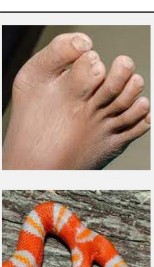

**Instruction:** What's abnormal in this image?
**LAMM's Response:**
The person in the image has a toe growth, which is a condition where an extra toe or a bony growth on the foot develops. This condition can be congenital, meaning it is present from birth, or it can develop later in life due to various causes.

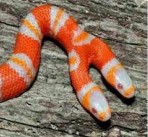

**Instruction:** What's abnormal in this image?
**LAMM's Response:**
The snake in the image has two heads. This is considered abnormal because snakes typically have a single head. The presence of two heads in this snake indicates a genetic mutation or some other form of abnormal development.

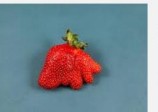

**Instruction:** What's abnormal in this image?
**LAMM's Response:**
The image shows a strawberry with a deformity, specifically a split in the fruit.

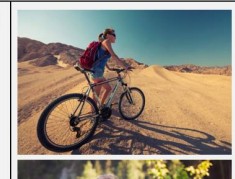

**Instruction:** What's the person in the picture doing?
**LAMM's Response:**
The person in the picture is a woman who is riding a bicycle. She is dressed in a blue tank top and blue shorts, and she is wearing sunglasses. She is also carrying a backpack, which suggests that she might be on a journey or a long-distance ride.

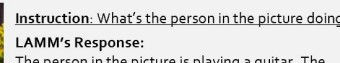

**Instruction:** What's the person in the picture doing?
**LAMM's Response:**
The person in the picture is playing a guitar. The reasoning process is based on the fact that the instrument being held in the hand is a guitar, and the person is actively engaged in playing it.

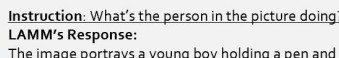

**Instruction:** What's the person in the picture doing?
**LAMM's Response:**
The image portrays a young boy holding a pen and practicing writing with a quill. He is sitting on a chair and appears to be focusing on his work. This activity indicates a keen interest or an education in the art of calligraphy, a traditional form of handwriting.

Figure 26: Example results of our model on in-the-wild images.

| Classification Question Pool |
| --- |
| 1.Can you identify the object in this image? |
| 2.Based on the image's features, what could be the potential category label for this image? |
| 3.What label would you assign to this image based on the object's shape and size? |
| 4.According to the model's prediction, what is the label assigned to this image? |
| 5.Can you provide the category label for this image based on the object's color and texture? |
| 6.What label do you think best describes the image's content? |
| 7.Based on the image's context, what category label would you assign to it? |
| 8.Can you suggest any alternate labels for this image based on its content and features? |
| 9.What is the most suitable category label for this image based on its shape, size, and context? |
| 10.According to the model's classification, what is the category label assigned to this object? |
| 11.Based on the image's visual cues, what category label do you think is the most appropriate? |
| 12.Can you provide any additional labels that could be applied to this image based on its context and features? |
| 13.What label would you assign to this image based on the object's function or purpose? |
| 14.According to the image's features and context, what label do you think best represents it? |
| 15.Can you suggest any potential alternate category labels that might be appropriate for this image based on its attributes? |
| 16.What is the most accurate category label for this image based on its features, context, and meaning? |
| 17.Based on the object's characteristics and the context of the image, what category label would you assign to it? |
| 18.According to the image's attributes, what label would you assign to it? |
| 19.Can you suggest any other category labels that could be applied to this image based on its features and meaning? |
| 20.What label do you think best represents the object in this image based on its shape, color, and texture? |
| **Classification Answer Pool** |
| 1.Upon close inspection of the image, it can be concluded that the majority of the objects in the image fall under the {C} category. |
| 2.Through extensive analysis of the image, it can be confidently stated that the image belongs to the {C} category. |
| 3.After careful examination of the image, it has been determined that the majority of the objects in the image can be classified as belonging to the {C} category. |
| 4.Based on a detailed examination of the image, it can be concluded that the image primarily consists of objects that fall under the {C} category. |
| 5.After closely analyzing the image, it has been determined that the main subject of the image belongs to the {C} category. |
| 6.Through rigorous examination of the image, it has been concluded that the image is primarily focused on objects that can be classified as belonging to the {C} category. |
| 7.Based on a thorough evaluation of the image, it can be confidently stated that the image is dominated by objects that fall under the {C} category. |
| 8.After careful scrutiny of the image, it has been determined that the majority of the objects in the image can be identified as belonging to the {C} category. |
| 9.Upon detailed analysis of the image, it can be concluded that the objects located at various positions in the image can be categorized as belonging to the {C} category. |
| 10.After thorough analysis of the image, it can be confidently stated that the image falls under the {C} category. |
| 11.Through careful scrutiny of the image, it can be confidently concluded that the image falls under the {C} category. |
| 12.After meticulous examination of the image, it has been determined that the image can be classified as {C}. |
| 13.After thorough inspection of the image, it can be confidently stated that the image falls under the {C} category. |
| 14.After extensive scrutiny of the image, it has been determined that the majority of the objects in the image belong to the {C} category. |
| 15.The image can be classified as {C} based on a close analysis of the objects and their characteristics. |

Figure 27: Question template pool and Answer template pool for classification task.

**Detection Question Pool**

1.What is the identity of the objects visible in the image, and where are they located?
2.Can you name each object in the image and describe its position accurately?
3.What objects are visible in the image, and where can they be found?
4.From the visual information provided, can you identify all the objects present in the image and describe their positions?
5.What are the names of the objects present in the image, and where are they positioned?
6.Can you accurately describe the location of each object visible in the picture?
7.What is the identity of each object in the image, and where can they be located?
8.From the visual clues, can you name and locate all the objects present in the picture?
9.Can you identify and describe the positions of all the objects visible in the image?
10.What objects can you see in the picture, and where are they placed?
11.Based on the context of the image, can you identify all the objects present and describe their locations?
12.Can you accurately report the names and positions of all the objects visible in the image?
13.What are the objects visible in the picture, and where can they be found?
14.From the visual information provided, can you name all the objects in the image and describe their positions accurately?
15.What is the name of each object present in the image, and what is its location?
16.Can you locate and identify all the objects in the image and describe their positions accurately?
17.What objects are present in the image, and where are they positioned relative to each other?
18.Based on the visual clues, can you name and locate all the objects visible in the image?
19.Can you identify all the objects present in the image and describe their relative positions?
20.What is the identity of the objects visible in the image, and how are they positioned?

**Detection Answer Pool**

1.An object that can be classified as {C} is located at the {P} position of the image.
2.The {C} object is present at the {P} coordinate in the image.
3.There is an object at the {P} location of the image that can be identified as belonging to the category of {C}.
4.An object categorized as {C} can be found at the {P} position in the image.
5.At the {P} position of the image, there is an item that falls under the category of {C}.
6.There exists an object categorized as {C} at the {P} position of the image.
7.The image contains an object that can be classified as {C} and is located at the {P} position.
8.You can spot an object belonging to the category of {C} at the {P} position in the image.
9.There is an object at the {P} position of the image, and its category is {C}.
10.Upon close inspection of the image, it can be observed that there is an object positioned at {P} that belongs to the {C} category.
11.At the exact coordinates of {P} in the image, there is an object that can be identified as belonging to the {C} category, and this object stands out from the rest of the objects in the image due to its unique color and pattern.
12.Scanning through the image, it becomes evident that there is an object at {P} that falls under the {C} category.
13.By carefully examining the image, one can spot an object at {P} that belongs to the {C} category.
14.Positioned at {P} within the image is an object that can be classified as belonging to the {C} category, and this object is also the only one in the image that has a specific type of texture and a distinctive shape that sets it apart from the other objects.
15.Upon careful examination of the image, it can be observed that there is an object positioned precisely at {P} that falls under the {C} category, and this object is also the only one in the image that has a specific type of pattern or design that makes it stand out from the rest of the objects.

Figure 28: Question template pool and Answer template pool for detection task in 2D vision.

| Keypoint Detection Question Pool |
|---|
| 1.Please locate the keypoints in the image and describe their position using xy coordinates.
2.Identify the keypoints in the image and describe their location using xy coordinates.
3.Please describe the location of the keypoints in the image using xy coordinates.
4.Identify the keypoints in the image and indicate their location using xy coordinates.
5.Please pinpoint the keypoints in the image and describe their position relative to each other using xy coordinates.
6.Locate the keypoints in the image and describe their position, size, and shape using xy coordinates.
7.Please describe the position of the keypoints in the image using xy coordinates and their visual features.
8.Identify the location of each keypoint in the image using xy coordinates and describe their visual characteristics.
9.Please locate and describe the position of all keypoints in the image using xy coordinates and their visual features.
10.Identify and describe the position of all keypoints in the image using xy coordinates and their visual characteristics.
11.Please describe the location and visual features of all keypoints in the image using xy coordinates.
12.Identify all keypoints in the image and describe their location, orientation, and visual characteristics using xy coordinates.
13.Please locate and describe the position and shape of all keypoints in the image using xy coordinates and their visual characteristics.
14.Identify the keypoints in the image using xy coordinates and describe their location in relation to the image edges and corners, as well as their visual characteristics.
15.Please describe the position and visual characteristics of all keypoints in the image using xy coordinates and their visual features. |
| **Keypoint Detection Answer Pool** |
| 1.The system has identified a keypoint at {P} in the image that can be classified as {C}.
2.There is a {C} keypoint located at the {P} position within the image according to the system's analysis.
3.The image contains a keypoint that can be classified as {C} at the {P} position.
4.The system has detected a {C} keypoint at the {P} coordinate in the image.
5.The {C} keypoint is present at the {P} position in the image, according to the system's analysis.
6.A keypoint that falls under the category of {C} is located at the {P} position of the image according to the system.
7.The system has identified a keypoint at the {P} position in the image that can be classified as {C}.
8.At the {P} coordinate in the image, the system has detected a keypoint that falls under the category of {C}.
9.The image contains a keypoint that can be classified as {C} at the {P} position according to the system's analysis.
10.The system has located a {C} keypoint at the {P} position within the image.
11.There is a {C} keypoint present at the {P} position in the image according to the system's analysis.
12.A keypoint that falls under the category of {C} has been identified at the {P} position of the image by the system.
13.The system has classified a keypoint at the {P} position in the image as {C}.
14.At the {P} coordinate in the image, the system has identified a keypoint that can be classified as {C}.
15.The image contains a {C} keypoint at the {P} position according to the system's analysis. |

Figure 29: Question template pool and Answer template pool for keypoint detection task in 2D vision.

| Counting Question Pool |
| --- |
| 1.Please count the number of objects that fall under a specific category in the image.
2.Can you identify the total number of instances of a certain class present in the image?
3.How many items in the image belong to a particular category?
4.Are you able to determine the exact count of objects that match a certain label in the image?
5.Please identify the number of objects that are categorized as a specific type and are present in the image.
6.Can you count the number of objects that share a common attribute in the image?
7.How many objects in the image can be classified under a certain category?
8.Please determine the quantity of objects in the image that belong to a specific class.
9.Can you identify the total number of objects with a certain label that are present in the image?
10.How many instances of a particular class can you pinpoint in the image?
11.Please count the number of objects in the image that are classified as a certain type.
12.Can you identify the number of objects in the image that match a specific category?
13.How many objects in the image fall under a certain classification?
14.Please determine the quantity of objects that belong to a specific category and are present in the image.
15.Can you count the number of items in the image that share a common feature or attribute?
16.How many objects in the image can be identified as a specific type?
17.Please identify the number of objects in the image that are labeled as a certain category.
18.Can you determine the total count of objects in the image that belong to a specific class?
19.How many instances of a certain class can you discern in the image?
20.Please count the number of objects in the image that have a specific label or category. |

| Counting Answer Pool |
| --- |
| 1.{N} {C} have been identified in the image according to the system.
2.The system has detected {N} {C} in the image.
3.The image contains {N} {C} according to the system's analysis.
4.{N} {C} have been detected within the image based on the system's analysis.
5.The system has identified {N} {C} present in the image.
6.There are {N} {C} visible in the image based on the system's analysis.
7.The image depicts {N} {C} according to the system.
8.{N} {C} have been identified within the image by the system.
9.The system has detected {N} {C} present in the image.
10.{N} {C} have been detected in the image according to the system's analysis.
11.There are {N} {C} present in the image based on the system's analysis.
12.The system has identified {N} {C} visible in the image.
13.The image contains {N} {C} as per the system's analysis.
14.{N} {C} have been detected within the image by the system's analysis.
15.The system has classified {N} {C} within the image. |

Figure 30: Question template pool and Answer template pool for counting task in 2D vision.

| 3D Object Classification Question Pool |
| --- |
| 1.How would you describe the point cloud in terms of its scenario?
2.What is the best scenario label for this point cloud based on the model's output?
3.What are some other possible scenarios that could explain this point cloud?
4.What scenario does this point cloud belong to according to the model's prediction?
5.What is the most accurate point cloud scenario label for this point cloud based on its features, context, and meaning?
6.Can you suggest any other scenario labels that could be applied to this point cloud based on its features and meaning?
7.According to the point cloud's attributes, what scenario label would you assign to it?
8.Based on the object's characteristics and the context of the point cloud, what point cloud scenario label would you assign to it?
9.What scenario label do you think best represents the object in this point cloud based on its features and objects?
10.Can you suggest any alternate scenario labels for this point cloud based on its content and features?
11.What is the most suitable scenario label for this point cloud based on its shape, size, and context?
12.According to the model's classification, what is the scenario label assigned to this point cloud?
13.Based on the point cloud's visual cues, what scenario label do you think is the most appropriate?
14.Can you provide any additional scenario labels that could be applied to this point cloud based on its context and features?
15.What scenario label would you assign to this point cloud based on the object's function or purpose? |

| 3D Object Classification Answer Pool |
| --- |
| 1. After conducting thorough analysis, it is evident that the point cloud in this scenario can be classified as {C}.
2. By carefully examining the data, it becomes clear that {C} is the most appropriate classification for this point cloud scenario.
3. Taking into account all the details, it can be determined that the point cloud falls under the scenario of {C}.
4. The analysis of this point cloud leads to the conclusion that it corresponds to the scenario of {C}.
5. Based on a comprehensive assessment, it is evident that the point cloud can be accurately categorized as a scenario of {C}.
6. This particular point cloud exhibits characteristics that align with the scenario of {C} upon closer examination.
7. Considering the available information, it can be confidently stated that this point cloud conforms to the scenario of {C}.
8. The features present in this point cloud indicate that it can be classified as a scenario of {C}.
9. Upon careful scrutiny, it is apparent that the point cloud fits the description of {C} scenario.
10. Analyzing the data within this point cloud leads to the identification of {C} as the most suitable scenario.
11. The observed attributes of the point cloud confirm that it corresponds to the scenario of {C}.
12. Taking into account the available evidence, it is evident that this point cloud scenario can be characterized as {C}.
13. Through meticulous analysis, it becomes evident that the point cloud aligns with the characteristics of {C} scenario.
14. By thoroughly examining the point cloud, it becomes clear that the scenario it represents can be labeled as {C}.
15. The properties and structure of this point cloud provide strong evidence that it corresponds to the scenario of {C}. |

Figure 31: Question template pool and Answer template pool for object classification in 3D vision.

| 3D Object Detection Question Pool |
|---|
| 1.What is the identity of the objects visible in the point cloud, and where are they located?
2.Can you identify and describe the positions of all objects visible in the point cloud?
3.Analyzing the point cloud, please list all objects present and specify where they are located.
4.Based on the visual data provided, name all the objects detected in the point cloud and describe their precise positions.
5.Can you accurately recognize and determine the locations of each object within the point cloud?
6.From the spatial information available, please identify all the objects present in the point cloud and provide details about their respective positions.
7.Describe the objects captured in the point cloud and outline their exact coordinates within the scene.
8.Based on the given point cloud, identify and label the objects visible, specifying their spatial placements.
9.Provide a detailed account of the objects observed in the point cloud, including their precise locations.
10.Analyze the point cloud and present a comprehensive inventory of the objects present, along with their respective positions.
11.Can you precisely name and locate all the objects detected within the point cloud based on the provided data?
12.Based on the spatial context of the point cloud, identify all the objects present and describe where they are situated.
13.Please identify and describe the positions of each object visible in the point cloud based on the available visual information.
14.Utilizing the spatial cues within the point cloud, can you accurately detect and outline the positions of all objects?
15.Based on the given point cloud, determine the objects contained within and provide an overview of their locations. |
| 3D Object Detection Answer Pool |
| 1.Positioned at the {P} location within the point cloud, an object within the {C} category can be observed.
2.By examining the {P} position of the point cloud, it is possible to identify an object categorized as {C}.
3.The point cloud includes an object at the {P} position, which can be classified under the category of {C}.
4.At the {P} position in the point cloud, there is an item categorized as {C}.
5.The {P} position of the point cloud allows for the identification of an object that belongs to the {C} category.
6.Within the point cloud, an object classified as {C} is situated at the {P} position.
7.An object that can be classified as {C} is located at the {P} position within the point cloud.
8.The {P} position of the point cloud reveals the presence of an object categorized as {C}.
9.At the {P} position within the point cloud, there exists an object that falls under the {C} category.
10.By examining the {P} position of the point cloud, it is evident that an object belonging to the {C} category is present.
11.The point cloud contains an object at the {P} position, which can be identified as {C}.
12.An object within the {C} category can be observed at the {P} position in the point cloud.
13.The {P} position of the point cloud displays the presence of an object that can be classified as {C}.
14.At the {P} position of the point cloud, an object can be identified as belonging to the {C} category.
15.Within the point cloud, an object categorized as {C} can be found at the {P} position. |

Figure 32: Question template pool and Answer template pool for object detection in 3D vision.

| 3D VQA Question Pool |
|---|
| 1.Please provide your responses to the following questions using the information depicted in the point cloud.
2.Based on the visual content of the scenario, please answer the following questions.
3.Utilizing the details presented in the point cloud, please respond to the following questions.
4.Analyze the visual elements of the point cloud and provide your answers to the following questions.
5.Without any additional context, use the point cloud provided to answer the following questions.
6.Your task is to examine the visual content of the point cloud and address the following questions.
7.Please utilize the details and visual cues depicted in the point cloud to answer the following questions.
8.Given the information conveyed in the point cloud, please provide your responses to the following questions.
9.Based on the visual data presented, respond to the following questions using the point cloud as your reference.
10.Analyze the content of the point cloud and provide your answers to the following questions.
11.Use the visual elements depicted in the point cloud to answer the following questions accurately.
12.Without any additional information, rely solely on the visual cues within the point cloud to address the following questions.
13.Please examine the visual content of the point cloud and provide your responses to the following questions.
14.Your task is to interpret the details and visual information in the point cloud to answer the following questions.
15.Utilize the information presented in the point cloud to formulate your answers to the following questions. |
| 3D VQA Answer Pool |
| 1.Upon careful examination of the point cloud, it becomes evident that the answer to this question is {A}.
2.By thoroughly analyzing the point cloud, one can reach the conclusion that {A} is the correct answer to this question.
3.The answer to this question can be deduced by closely observing the details within the point cloud, leading to the determination that {A} is the answer.
4.If you closely inspect the point cloud, you will discover that the answer to this question is {A}.
5.Based on the analysis of the point cloud, it can be confidently stated that the answer to this question is {A}.
6.The examination of the point cloud provides a clear indication that {A} is the answer to this question.
7.By closely studying the point cloud, one can ascertain that the correct answer to this question is {A}.
8.After careful analysis of the point cloud, it is evident that {A} is the answer to this question.
9.The answer to this question can be determined by carefully examining the details present in the point cloud, resulting in the conclusion that {A} is the answer.
10.Through a meticulous analysis of the point cloud, it becomes apparent that the answer to this question is {A}.
11.Upon thorough examination of the point cloud, it can be inferred that the answer to this question is {A}.
12.By closely observing the point cloud, one can derive that the answer to this question is {A}.
13.After a detailed analysis of the point cloud, it is evident that the correct answer to this question is {A}.
14.The answer to this question can be revealed by closely inspecting the point cloud, and it is {A}.
15.By carefully scrutinizing the details in the point cloud, it can be concluded that {A} is the answer to this question. |

Figure 33: Question template pool and Answer template pool for visual question answering in 3D vision.

