# OpenReview forum: "LAMM: Language-Assisted Multi-Modal Instruction-Tuning Dataset, Framework, and Benchmark"
_NeurIPS.cc/2023/Track/Datasets_and_Benchmarks — NeurIPS 2023 Datasets and Benchmarks Poster_

### Official Review · Reviewer_Byte · 2023-07-12
**The paper is well-written. The contributions are clear.**

**Rating:** 8
**Confidence:** 3
**Correctness:** Yes. The dataset is constructed in a …
**Clarity:** Yes

**Strengths:**

LAMM-Dataset is the first instruction tuning datasets for point cloud, and the LAMM-Benchmark is the first attempt of a benchmark for MLLMs that offers a comprehensive evaluation of existing models on various computer vision tasks.

**Additional Feedback:**

No

**Documentation:**

Yes

**Limitations:**

The scale of point cloud dataset could be enlarged.

**Opportunities For Improvement:**

The size of public available datasets used to constitute LAMM-Dataset is not introduced, and will this affect the performance of different tasks?

**Relation To Prior Work:**

Yes. Within the introduction, the authors discussed the advantages of the proposed LAMM-Dataset and LAMM-Benchmark.

**Summary And Contributions:**

This paper introduces LAMM-Dataset and LAMM-Benchmark, which extend the research of MLLMs to point clouds. Experiments validate the effectiveness of the dataset and benchmark. The methods of constructing the dataset and benchmark are demonstrated in detaile. A primary MLLM training framework is also proposed.

---

> ### Author Response · Authors · 2023-08-19
>
> Thank you for your valuable feedback. We have carefully analyzed your comments and made improvements in our rebuttal. Here is the revised response to the issues you raised
>
> **Regarding the size of publicly available datasets:**
>
> We would like to provide further clarification on the composition of the LAMM-Dataset. To clarify, all the image and point cloud data in the LAMM-Dataset are sourced from publicly available datasets. Specifically, for LAMM-2D-Dataset, 49k COCO images were used to generate the Daily Dialogue & Detailed Description training data. Additionally, 42k Bamboo images were utilized to generate the Factual Knowledge Dialogue data. Furthermore, approximately 46k images were randomly sampled from datasets such as TextVQA, LocoCount, COCO Keypoints, and Bamboo to generate Visual Task Dialogues. Regarding LAMM-3D-Dataset, the sample sizes for each dataset are as follows: 2k ShapeNet, 1.3k 3RScan detection, 2k CLEVR3D, and 4.9k GPT-generated dialogue samples. For more detailed information, please refer to Appendix Section B of our submission. Besides, We also conducted comprehensive experiments on the LAMM-Dataset to explore the relationship between dataset size and task performance. Interestingly, our findings indicate that the size of the public dataset has a relatively limited impact on task performance. These results align with the insightful observations in the LIMA[1] paper, which emphasizes that large language models primarily acquire their knowledge during the pre-training phase. The paper also highlights that fine-tuning with a limited amount of data can effectively guide the model to produce high-quality outputs.
>
> **Regarding the scale of the point cloud dataset:**
>
> We greatly appreciate your suggestion, and indeed, our point cloud dataset can be expanded. However, it's worth noting that annotating 3D datasets can be costly, and currently, mainstream 3D datasets tend to be relatively small. For instance, the ScanNet dataset, which focuses on indoor scenes, consists of only 1,513 scenes. Therefore, the size of the point cloud data in our dataset is already significant within the context of point cloud datasets. However, we can gradually expand the point cloud portion of our dataset in the future.
>
> Thank you once again for your valuable input, and we have incorporated the improvements into our rebuttal. If you have any further questions or need additional discussion, please feel free to let us know.
>
> [1] Zhou C, Liu P, Xu P, et al. Lima: Less is more for alignment[J]. arXiv preprint arXiv:2305.11206, 2023.

---

### Official Review · Reviewer_FCRL · 2023-07-21
**A high-quality multimodal dataset based on gpt-4**

**Rating:** 5
**Confidence:** 4
**Correctness:** Yes.
**Clarity:** Yes.

**Strengths:**

1. This work provides high-quality human-gpt dialogues on multi modal examples that could improve the cross-modality unification and performance of human-machine dialogue system and robots in real-scenery.
2. The authors implmented a wide range of experiments and comparison across various vision-language tasks that help the community to sense the newest improvements on this area.
3. the newly-included point cloud field could guide large models towards new applications.

**Additional Feedback:**

Nothing yet.

**Documentation:**

dataset details are well curated.

access: github. **** you guys have leaked the information of the authors via github repo****. I wonder if this is appropriate.

license: The LAMM-Dataset is a compilation of publicly available datasets that have been licensed under the Creative Commons license (CC-BY)

**Ethics:**

Ethical concerns: how did you "ensure apporiate content"(mentioned in the appendix). You need to convince me that you've implemented strict correlation-filtering or human-supervision.

**Limitations:**

Yes, the authors addressed their work's failure in certain tasks and showed the model's performance on various tasks.

**Opportunities For Improvement:**

1. The size of this dataset seems not large enough.
"186,098 image-language instruction-response pairs and a point cloud instruction-tuning dataset with 10,262 point cloud-language instruction-response pairs."

I understand that point cloud data is difficult to collect.
However, since your image data is straightly accessed through public available datasets, I think 186,098 is a relatively small size in this domain.


2. Also, how did you select your data from other datasets? This is an important process of constructing a dataset.
I believe I didn't see this clearly from your essay.


3*. One more thing for your good is that maybe you can deploy a LLM locally and fine-tune it instead of simply calling a web GPT-API. This could help your work become transparent and independent and your whole construction process could be completive.

**Relation To Prior Work:**

Yes.

**Summary And Contributions:**

Summary and contributions:
1.  Involve point-cloud modality in this cross-modal dataset.
2.  Add high-quality and practical knowledge to existing public available datasets by instructing GPT-4.
3.  Comprehensive comparison with prevalent related works that demostrates the quality of this dataset.
4.  A framework that helps the community to utilize this dataset.

---

> ### Author Response · Authors · 2023-08-19
>
> We would like to express our sincere gratitude for your valuable feedback and constructive criticisms on our submitted paper. We have carefully reviewed your comments and in the following sections, we address each point individually, providing comprehensive explanations and discussions.
>
> **About the size of the dataset:**
>
> While it is true that the dataset size of 186,098 image-language instruction-response pairs may appear relatively small in comparison to some other domains, we have conducted experiments that demonstrate the significance of data quality over data quantity. As shown in Figure.5(d) in manuscript, we evaluate models on ScienceQA test set. With only daily dialogue and detailed description, models trained on 10K samples obtain an accuracy of 39.21%, while 98K samples boost performance to 39.46% and LLaVA obtains 40.5%. However, 42K samples of factual knowledge dialogue data can improve accuracy from 39.46% to 43.82%, which also surpass results of LLaVA with similar quantity. And the addition of 46K visual task dialogue samples can further improve the result to 47.09%.
>
> Our experimental results indicate that having a higher quality dataset, even if it is smaller in size, can lead to better performance compared to having a larger dataset with lower quality. This aligns with related work in NLP, such as the conclusions from LIMA[1], which also support the notion that data quality is more important than sheer data quantity.
>
> Although the image data was collected from publicly available datasets, the process of curating and aligning the data with language instructions involves careful selection and validation to ensure the dataset's quality and relevance to the task. By prioritizing data quality, we aim to provide a more reliable and effective training and evaluation environment for MLLMs. We believe that our focus on quality over quantity contributes to the overall validity and credibility of our dataset.
>
> **About the data selection:**
>
> We apologize for the lack of clarity in the manuscript. A more detailed explanation of our data selection process is provided as following:
>
> The dataset consists of four types of data. The "daily dialogue" and "description" data were sampled from the COCO images in Visual Genome dataset, which comprises approximately 49k images. We collected both conversation and description data specifically from this subset of images.
>
> For the "factual knowledge dialogue" data, we utilized the description data from the Bamboo dataset and Wikipedia descriptions to generate a total of 42k instances. These instances were generated using a combination of the Bamboo dataset and Wikipedia descriptions.
>
> Lastly, the data related to computer vision tasks were also created in a similar manner. We generated approximately 46k instances for each specific computer vision task, resulting in a dataset of approximately 10k instances for each task.
>
> More detailed information on data selection will be integreted in the final version of our manuscript.
>
> **About the usage of web GPT-API:**
>
> Our research focuses on the utilization of GPT-4 for data generation, which offers superior quality compared to other smaller language models. As shown on the Opencompass [2] leaderboard, the performance of GPT-4 outperforms other models across various metrics.
>
> Regarding the suggestion of fine-tuning on Llama, we have thoroughly considered the possibility. However, the challenge lies in acquiring suitable data for fine-tuning the Llama model. Given the nature of our task, obtaining high-quality, domain-specific data for fine-tuning purposes remains a significant hurdle. Therefore, we have chosen to leverage the powerful capabilities of GPT-4 for data generation, as it provides a robust and effective solution for our research objectives.
>
> **About the documentation:**
>
> We have taken note of the reviewer's concern regarding the disclosure of authors' information through the GitHub repository. In accordance with the requirements of NeurIPS, we submitted our manuscript under a double-blind setting. Following the guidelines of the NeurIPS Datasets and Benchmarks track, we promptly made our data, code, and models publicly available immediately after submission, with the project homepage link provided on OpenReview.
>
> Adhering to the customary practice of open sourcing, our project homepage does contain relevant organizational information. However, upon reviewing the feedback, we promptly took action to address the raised concern by removing the authors' information from the GitHub repository. We sincerely hope that this action has not had any adverse effects on our submission.
>
> [1] LIMA: Less Is More for Alignment
>
> [2] https://opencompass.org.cn/leaderboard-llm

---

> > ### Author Response · Authors · 2023-08-19
> >
> > **About ethics:**
> >
> > We appreciate the recognition of the ethical concerns related to the dataset. We manually curated and reviewed the data to ensure the ethical considerations in dataset creation.
> >
> > We understand that automating the screening process can be an area of improvement, and we are actively exploring and researching automated methods for data alignment. While alignment remains an unsolved field, we are committed to staying updated on the latest advancements and developments in this area.
> >
> > Additionally, we recognize the importance of ongoing monitoring and follow-up to address any ethical issues that may arise. We are dedicated to maintaining a vigilant approach and continuously refining our practices in line with evolving ethical standards.
> >
> > We sincerely appreciate your guidance and attention. We look forward to receiving your evaluation of the revised version of our paper and hope that our work meets the high standards and requirements.

---

### Official Review · Reviewer_XzFC · 2023-07-21
**Review of LAMM, Submission 107**

**Rating:** 5
**Confidence:** 3

**Strengths:**

- The paper tackles a popular and interesting subject: adapting LLMs for various visual tasks and modalities. This is worth exploring, given the impressive performance that LLMs have demonstrated in various NLP tasks, and also given the large number of high-quality LLMs that are becoming easily accessible.
- The paper introduces a new dataset that covers many visual tasks, both 2D and 3D.
- The paper presents a comprehensive evaluation of the presented benchmark.
- The paper is well-written and easy to follow. The figures are easy to understand and detail the dataset creation process and benchmark evaluation pipeline.


**Additional Feedback:**

No additional feedback.


**Clarity:**

As mentioned in the Strengths section, the paper is well explained and easily comprehended.


**Correctness:**

As mentioned in the Strengths section, the benchmark construction is well explained, the authors also provide implementation code to download the dataset and reproduce the results.


**Documentation:**

As mentioned in the Strengths section, the benchmark construction is well explained, the authors also provide implementation code to download the dataset and reproduce the results.


**Ethics:**

Since the proposed dataset of instruction-response pairs was constructed using GPT-4, this might raise some concerns. Firstly, using GPT-4 for data annotations might be against the product's terms of service. Secondly, utilizing GPT-4, which was pre-trained on unknown data, might also raise concerns regarding data leakage or the legitimacy of such data. Overall, the use of GPT-4 might limit the application of this dataset beyond research.

**Limitations:**

- Novelty: While introducing 3D tasks and grounding using different targets when generating the instructions are novel contributions, overall, both the dataset construction method and the model used follow standard and popular methods for data annotation and LLM adaptation to visual tasks and other modalities. For instance: LLM generated instruction data: [1, 2], grounding using different ground truths like boxes and captions [3], and using a linear layer to project the different modalities into the LLMs embeddings space: [4, 5]
- Dataset: The authors introduce a set of design choices to construct the dataset, using different tasks and then prompting GPT4 using a set of templates and prompts to construct the instruction-response pairs. However, the authors did not provide any arguments for design decisions or some comparison using various inputs to ground the answers for GPT4, for instance. This is mainly a question due to the importance of designing templates and prompts for LLMs.
- Evaluation pipeline: The evaluation pipeline consists of converting the input into an instruction format, using the LLM to generate the output in a human language form, and then extracting output entities for task-specific evaluation. However, it is well-documented [6,7] that for localization and fine-grained tasks like detection, segmentation, and counting require task-specific tokens (horizontal and vertical bins for predicting the bounding box coordinates for example) is necessary to obtain good results, and casting the output as standard output (floating points for detection) does not yield good results. This is demonstrated in the obtained results where the model struggles on most tasks (detection, key points, counting), even with fine-tuning.
- Metrics: While the BLEU score might not be an appropriate metric, the usage of a GPT-based metric can also raise many questions: stability and reproducibility; GPT models are stochastic, and the model endpoints constantly change. Self-performance bias; the model is trained on GPT-generated targets, and this might add some performance bias.
- Model performance: Overall, the performance of the proposed LAMM is mixed, not on par with other MLLM methods like LLaVA, and quite far away from being applicable given their gap to task-specific models.

[1] Self-instruct: Aligning language model with self-generated instructions

[2] Instruction tuning with GPT-4

[3] Visual Instruction Tuning

[4] Linearly Mapping from Image to Text Space

[5] Enhancing Vision-language Understanding with Advanced Large Language Models

[6] Pix2seq: A Language Modeling Framework for Object Detection

[7] A Study of Autoregressive Decoders for Multi-Tasking in Computer Vision


**Opportunities For Improvement:**

See the Limitations section.


**Relation To Prior Work:**

Overall, the related work section is well-written and refers to various related works.

**Summary And Contributions:**

The paper introduces a new dataset & benchmark for evaluating multi-modal LLMs (MLLMs) on 2D and 3D visual tasks, which include object detection, classification, key-point detection, object counting, OCR, and 3D indoor detection.

First, the multi-modal dataset, named LAMM-Dataset, is constructed using eight image datasets and four-point cloud datasets: Bamboo (classification and detection), COCO (captions, detection, and key-points), Visual Genome, Locount, Text VQA, 3RScan, CLEVR 3D, 3D SSG, and ShapeNet. The LAMM-Dataset consists of instruction-response pairs, which are constructed by prompting the GPT4 API using the original labels from each dataset and a set of pre-defined prompts that are task-specific.

Then, the authors introduce the LAMM-Benchmark, an appropriate benchmark for evaluating MLLMs. The benchmark introduces an inference pipeline for processing the input instructions and extracting output entities to ensure that the MLLMs produce reasonable task and dataset-specific outputs that can be used for computing the appropriate metrics and evaluating the models.

Finally, the authors introduce the LAMM-Framework that consists of adapting pretrained and modality-specific encoders and an LLM used to solve the tasks presented in the LAMM-Dataset, utilizing the pipeline and metrics of the LAMM-Benchmark.

---

> ### Author Response · Authors · 2023-08-19
>
> We would like to express our sincere gratitude for your valuable feedback and constructive criticisms of our submitted paper. We have carefully reviewed your comments and in the following sections, we address each point, providing comprehensive explanations and discussions.
>
> **About novelty:**
>
> In designing the framework for our baseline model and utilizing GPT-API for data generation, we drew inspiration from existing works. However, our key contribution lies in providing the open-source community with a comprehensive pipeline. This pipeline involves generating multi-modal instruction tuning data, training a large-scale multi-modal language model, and evaluating the model on visual and multi-modal tasks. We have made all the corresponding data, code, and models publicly available.
>
> **About dataset:**
>
> In both dataset generation and benchmark evaluation, we have utilized the GPT-API. During dataset generation, we also ablation on different prompts for GPT-API, manually observing their reasonableness. Eventually, we selected a suitable prompt for batch generation. Specifically, our focus is not primarily on the impact of different prompts on the quality of the generated text. Instead, we emphasize the influence of different types of datasets, which contain varying data distributions, on instruction tuning for the model.
>
> **About the evaluation pipeline:**
>
> Indeed, current models face challenges in handling tasks involving the generation of floating-point numbers, such as object detection and keypoint detection. Some existing approaches [1] address this issue by using special tokens, while others [2] achieve good results by purely generating textual descriptions. Regardless of whether these models employ special tokens, the output they generate is in the form of textual descriptions. As for prompt input for the evaluation of different models, we followed prompts used by Vicuna [3] and LLaVA [4] in their work. In our evaluation, we adopt a unified text-based evaluation approach to assess these models. This choice allows us to compare and evaluate the performance of different models consistently, regardless of the specific techniques they employ.
>
> **About metrics:**
>
> Although the use of GPT for evaluation in multimodal tasks is still in its early research stage, there have been significant advancements in employing GPT for evaluation in the field of NLP [5]. These studies have demonstrated a certain level of stability and reproducibility in their evaluations using GPT.
>
> Given this progress, we have adopted a similar approach in GPT evaluation for multimodal tasks by following the methodologies established in NLP research. We aim to leverage the expertise and insights gained from NLP evaluation using GPT and apply them to the evaluation of GPT in multimodal tasks.
>
> For self-performance bias, we align with the work of LLaVA [2], where we generate data using GPT. We acknowledge that the issue of self-performance bias is currently unavoidable. We understand the importance of addressing this concern and are committed to continuously monitoring and addressing any potential biases in our evaluation process.
>
> **About model performance:**
>
> It is important to note that the primary objective of our study was not to outperform task-specific models across all datasets but rather to explore the effectiveness of training data and assess the potential of zero-shot learning in MLLM. As most tasks do not yet have zero-shot learning results for task-specific models available, we provided supervised training results as a top-borderline.
>
> Our intention is to establish a starting point and demonstrate the efficacy of training data in the context of MLLM. While our model may not currently outperform all other MLLMs like LLaVA, we believe this work contributes to the understanding of data effectiveness and paves the way for future improvements and advancements in MLLM research.
>
> [1] KOSMOS-2: Grounding Multimodal Large Language Models to the World
>
> [2] Shikra: Unleashing Multimodal LLM's Referential Dialogue Magic
>
> [3] Vicuna: An Open-Source Chatbot Impressing GPT-4 with 90%* ChatGPT Quality
>
> [4] Visual Instruction Tuning
>
> [5] AlpacaEval : An Automatic Evaluator for Instruction-following Language Models

---

> > ### Author Response · Authors · 2023-08-19
> >
> > **About ethics:**
> >
> > OpenAI does not claim copyright over content generated by the API for users or their end users [6]. This alleviates concerns regarding ownership and usage rights of the generated content. Furthermore, it is claimed that the data used to train models at OpenAI is obtained lawfully [7], which helps address concerns related to data leakage and the legitimacy of the training data. This emphasis on ethical data practices is crucial in ensuring the trustworthiness and integrity of the models and their outputs. As users of the GPT-API, we find reassurance in knowing that these issues are duly addressed and taken seriously.
> >
> > [6] https://openai.com/policies/terms-of-use
> >
> > [7] https://help.openai.com/en/articles/7842364-how-chatgpt-and-our-language-models-are-developed
> >
> > We sincerely appreciate your guidance and attention. We look forward to receiving your evaluation of the revised version of our paper and hope that our work meets the high standards and requirements.

---

### Official Review · Reviewer_SwmA · 2023-07-26

**Rating:** 7
**Confidence:** 3
**Correctness:** Everything seems okay to me.

**Strengths:**

- The paper makes an important contribution in the form of an instruction-tuning dataset that also includes point cloud tasks as an additional visual modality.

- The construction of the LAMM dataset is very well done, going beyond the simple technique of adding an instruction to standard multimodal tasks by constructing multi-round dialog instruction-response pairs.

- The benchmark design seems good, albeit standard. Adding the output format as part of the input makes a lot of sense. Having GPT preference ranking as part of the benchmark is a nice addition.

- The experiments and analysis seem fairly thorough. The authors do a good job of analyzing their various results.

- The paper is well structured and well written, especially considering the volume of work that has to be presented.

Overall the paper makes several important contributions, and includes all the required experiments that would be expected, which is not easy to cram into 9 pages.

**Additional Feedback:**

-

**Clarity:**

Overall clarity looks good -- please refer to last point in "Opportunities for Improvement"

**Documentation:**

Yes. Some details on how much it cost to collect the data using the GPT API would be interesting.

**Limitations:**

No discussion of limitations.

**Opportunities For Improvement:**

- The authors list several multimodal instruction tuning datasets in Section 2, but do not distinguish how their LAMM dataset differs from and improves upon the other datasets, or what gap their dataset fills that the others miss.

- The Binary Location Metric is not well motivated -- why introduce a metric specifically for object localization? Why is this task important enough to warrant its own evaluation?

- More models could be used for the GPT eval, beyond just LLaVA and the LAMM tuned model.

- The LAMM acronym is used to mean many different things throughout -- sometimes it's the instruction tuning data, sometimes it's the benchmark, sometimes it;s the model trained on the data. I would suggest making the terminology more clear (e.g. Vicuna-LAMM for the model trained on the LAMM data)


**Relation To Prior Work:**

Please refer to first point in "Opportunities for Improvement"

**Summary And Contributions:**

The authors introduce LAMM, an instruction-tuning dataset and benchmark for 2D vision and 3D point cloud understanding tasks.

- The LAMM dataset uses the GPT API to convert multimodal task inputs into instruction-response pairs, either in the form of multi-round dialogues or single-round tasks. The LAMM dataset consists 8 diverse 2D vision tasks and 4 point cloud datasets.

- The LAMM benchmark evaluates models on a separate set of 9 2D vision tasks and 3 point cloud tasks. Instruction-tuned models are evaluated in a zero-shot manner by providing a task definition, output format, and query. For evaluation, the model uses both the task-specific metric and a Binary Locating Metric for object recognition. Additionally, GPT is used to evaluate models by ranking different models' outputs.

- The LAMM training framework optimizes feature projection layers and LoRA during training, while keeping the vision (2D image and 3D point cloud) encoders and language backbone (Vicuna-13B) frozen.

---

> ### Author Response · Authors · 2023-08-19
>
> Thank you for the feedback. We have carefully reviewed the issues raised and refined our responses accordingly:
>
> **Compared with other MLLM datasets:**
>
> 1) In contrast to the pure text instruction fine-tuning datasets like FLAN, ShareGPT, and Alpaca, our innovative LAMM dataset takes a step further by embracing multimodal instruction fine-tuning. While other multimodal datasets such as Multi-Instruct, Mini-GPT4, LLaVA, and Otter typically incorporate only image and text modalities, our LAMM dataset goes beyond and integrates the powerful point cloud modality.
> 2) What sets our LAMM dataset apart is its core objective of endowing large language models with the ability to tackle traditional computer vision tasks. Our dataset construction pipeline revolves around traditional 2D and 3D datasets, meticulously transformed into a format that can be effectively trained and evaluated by LLM models. By incorporating this multimodal approach and enabling LLM models to handle computer vision tasks, our LAMM dataset stands out as a unique and valuable resource in the field.
>
> In summary, the LAMM dataset represents a significant advancement in multimodal instruction fine-tuning. Its incorporation of multiple modalities, including text, images, and point clouds, coupled with its emphasis on empowering large language models to address traditional computer vision tasks, distinguishes it from other datasets in the field.
>
> **Binary Location Metric:**
>
> The Binary Location Metric introduced in LAMM-Benchmark provides valuable insights into the localization abilities of LLM models. While computer vision tasks may not be the ideal benchmark for MLLM, this metric offers a novel approach to evaluate object localization in LLM models. It addresses the challenge of accurately outputting object bounding box information in purely text-based generative interactions. The Binary Location Metric serves as a valuable tool to assess and understand the current state of localization in LLM models.
>
> **Evaluating more models with GPT eval:**
>
> We appreciate the reviewer's suggestion regarding the inclusion of additional models for the evaluation of GPT. However, due to the limitations imposed by the use of GPT-4 in our study, we initially focused on comparing LLaVA[2] and our own model. Nevertheless, in response to the reviewer's comment, we have extended our experimental analysis to include the other two models inferred in our work: MiniGPT-4[3] and mPLUG-owl[4].
>
> To ensure a fair and consistent evaluation, we have employed a direct scoring approach for the extended evaluation. Following MM-Vet[1], We present the results by assigning scores to the outputs generated by each model. The scoring criteria were based on accuracy, relevance, fluency, logical coherence, and information richness. Note that the evaluation pipeline and data change a lot, so the results differ from the original result provided in the manuscript. We will continually update these results in our leaderboard both in the revised version of the manuscript and our homepage. The results of this extended evaluation are as follows:
>
> - | MLLMs           | LLaVA | MiniGPT | mPLUG-owl | LAMM  |
>   | --------------- | ----- | ------- | --------- | ----- |
>   | GPT Eval. Score | 50.16 | 7.28    | 41.88     | 48.44 |
>
> **Naming convention:**
>
> We will promptly address any ambiguity in the naming convention to ensure clarity and accuracy in our final submission. We appreciate your attention to detail and will make the necessary revisions accordingly.
>
> If you have any further questions or require additional information, please let us know. We will incorporate these refinements into our rebuttal.
>
> [1] MM-Vet: Evaluating Large Multimodal Models for Integrated Capabilities
>
> [2] Visual instruction tuning
>
> [3] MiniGPT-4: Enhancing Vision-Language Understanding with Advanced Large Language Models
>
> [4] mPLUG-Owl : Modularization Empowers Large Language Models with Multimodality

---

### Official Review · Reviewer_8BVz · 2023-07-28
**A valuable and timely benchmark, but uses LLM-generated text.**

**Rating:** 4
**Confidence:** 2
**Correctness:** The claims appear to be correct.

**Strengths:**

The paper has many strengths:
- There is a need for this kind of benchmark at the moment -- the work is timely.
- For a fine-tuning dataset, the dataset is still reasonably large, with almost 200k language-image observations.
- The paper introduces the first ever benchmark for MLLMs
- The authors include a default training pipeline for MLLMs with their benchmark, which is useful for future work in this area
- The paper includes a fairly thorough evaluation of different methods.

**Additional Feedback:**

Overall the main thing that seems missing from this work is the acknowledgment and discussion of the changing practices involving language models, particularly as it pertains to generating datasets and evaluation. On the whole, I think that it is arguably reasonable to include LLM-generated data and evaluations using a downstream LLM, but it is precisely that -- arguable. The authors should include a discussion of this, as this is an important component of this type of work. If the authors can address this concern (mainly about using LLM-generated data), then I will raise my score.

**Clarity:**

The paper is clear, although the inclusion of both a LAMM-Dataset and LAMM-Benchmark is somewhat confusing, with respect to the naming.

**Documentation:**

Yes, the included documentation appears to be fine. I also liked the video on dataset website.

**Ethics:**

None.

**Limitations:**

The authors do not discuss the limitations of this work, but they are upfront about this in the author checklist. In my opinion, any work related to instruction tuning (and especially most anything involving LLM usage) should discuss its potential for negative social impact. I think that a discussion of this should be included somewhere in the text.

**Opportunities For Improvement:**

- At least a portion of the text in the dataset is generated using the GPT API -- can the authors clarify what exactly they mean by this? Is the data generated using ChatGPT? If so, this seems like a fairly significant downside to the dataset. If the data itself is generated by a model, this should be discussed as a limitation of the work, at a minimum.
- One of the evaluation metrics that is used is the GPT score. In my opinion, this is fine, as long as other metrics are included as well (they are), but this should also be discussed in the limitations.

**Relation To Prior Work:**

The paper discusses the relation to prior work.

**Summary And Contributions:**

This paper introduces a benchmark for performing instruction fine tuning of multimodal LLMs (MLLMs) called LAMM. LAMM extends prior work that focuses on text and images, to additionally include modalities such as point clouds and additional vision tasks. The authors additionally analyze existing methods, and show that their own method, as evaluated on their benchmark, outperforms prior work.

---

> ### Author Response · Authors · 2023-08-19
>
> Thank you for your feedback regarding our proposed use of the GPT-API. We appreciate the opportunity to clarify the confusion about the GPT-API and discuss the limitations associated with data generated by LLMs. We understand the importance of ongoing research and development to mitigate the limitations and are committed to refining our techniques and collaborating with the research community to enhance the performance and address the identified limitations. Related discussions will be integrated into the revised version later.
>
> **How to use GPT-API:**
>
> Firstly, we apologize for any misunderstanding caused. In our study, we employed GPT-API, the state-of-the-art language model, to generate the multi-modal instruction data. To generate data in the desired format, including multi-round dialogue and one-round detailed descriptions, we provided system messages and example dialogues to guide the generation process of GPT-API. The use of GPT-API for generating text-based conversations has been widely adopted in Natural Language Processing or multi-modal data [1, 2, 3] has shown promising results in various tasks.
>
> **Limitations of GPT-generated data:**
>
> Even though GPT-API provides a convenient interface to access the model's capabilities, it does not change the fundamental characteristics or limitations of the underlying GPT model. Some of the limitations we acknowledge include:
>
> **(1) Data noise:** GPT-API has no direct access to visual information and only textual context such as captions, attributes can serve as its information source, restricting its understanding of images and resulting in missing detailed information.
>
> **(2) GPT-API incapacity:** While GPT-API excels at generating coherent and contextually relevant responses, it occasionally generates responses that may appear plausible but are factually incorrect or lack context. And it sometimes struggles with understanding complex or ambiguous queries. Meanwhile, generated data used for training may inadvertently reflect inherent biases of GPT-API [4].
>
> **Limitation mitigation:** We have taken several measures in our research, such as carefully crafting prompts that incorporate relevant image information and using prompts strategically to guide the model's responses (See Section 1 in the Appendix). Furthermore, we perform manual checks on the generated data to identify and filter out low-quality data or any instances that exhibit significant errors or biases. This manual review process helps ensure the quality and suitability of the generated conversations for instruction tuning. A demonstration of generated-data quality will be integrated into the final vision of our paper as well.
>
> [1] Hajishirzi. 2023. Self-Instruct: Aligning Language Models with Self-Generated Instructions. In Proceedings of the 61st Annual Meeting of the Association for Computational Linguistics (Volume 1: Long Papers), pages 13484–13508, Toronto, Canada. Association for Computational Linguistics.
>
> [2] Rohan Taori, Ishaan Gulrajani, Tianyi Zhang, Yann Dubois, Xuechen Li, Carlos Guestrin, Percy Liang, and Tatsunori B. Hashimoto. Stanford alpaca: An instruction-following llama model.
>
> [3] Haotian Liu, Chunyuan Li, Qingyang Wu, and Yong Jae Lee. Visual instruction tuning, 2023.
>
> [4] OpenAI. Gpt-4 technical report. ArXiv, abs/2303.08774, 2023.

---

### Decision · Program_Chairs · 2023-09-22

**Decision:**

Accept (Poster)

**Comment:**

Overall half the reviewers are in favor of this paper and the responses to the others seem to have addressed their concerns. I am in favor of accepting this paper.